# Inferring brain plasticity rule under long-term stimulation with structured recurrent dynamics

**Zhichao Liang**[1,2*], **Jingzhe Lin**[1*], **Xinyi Li**[1*], **Guanyi Zhao**[1], **Quanying Liu**[1,3†]

1. Department of Biomedical Engineering, Southern University of Science and Technology
2. Center for Neurocognition and Social Behavior, Artificial Intelligence Research Institute, Shenzhen University of Advanced Technology
3. Shenzhen Loop Area Institute
* These authors contributed equally.    † Corresponding author: liuqy@sustech.edu.cn

## Abstract

Understanding how long-term stimulation reshapes neural circuits requires uncovering the rules of brain plasticity. While short-term synaptic modifications have been extensively characterized, the principles that drive circuit-level reorganization across hours to weeks remain unknown. Here, we formalize these principles as a latent dynamical law that governs how recurrent connectivity evolves under repeated interventions. To capture this law, we introduce the Stimulus-Evoked Evolution Recurrent dynamics (STEER) framework, a dual-timescale model that disentangles fast neural activity from slow plastic changes. STEER represents plasticity as low-dimensional latent coefficients evolving under a learnable recurrence, enabling testable inference of plasticity rules rather than absorbing them into black-box parameters. We validate STEER with four benchmarks: synthetic Lorenz systems with controlled parameter shifts, BCM-based networks with biologically grounded plasticity, a task learning setting with adaptively optimized external stimulation and longitudinal recordings from Parkinsonian rats receiving closed-loop DBS. Our results demonstrate that STEER recovers interpretable update equations, predicts network adaptation under unseen stimulation schedules, and supports the design of improved intervention protocols. By elevating long-term plasticity from a hidden confound to an identifiable dynamical object, STEER provides a data-driven foundation for both mechanistic insight and principled optimization of brain stimulation. The source code of this study is available at `https://github.com/ncclab-sustech/STEER.git`.

## 1 Introduction

Understanding long-horizon plasticity, the principles by which neural circuits reorganize over days to weeks, is a central problem in neuroscience and a broader challenge for learning systems (Turrigiano, 2012; Abraham et al., 2019; Appelbaum et al., 2023). These rules determine how the brain adapts during learning and memory (Speranza et al., 2021) and how interventions reshape network function over repeated sessions (Huang et al., 2005; Suppa et al., 2016; Sandoval-Pistorius et al., 2023). Yet despite decades of study, we still lack a predictive rule that connects repeated stimulation to slow circuit reconfiguration. Without such a rule, we cannot anticipate how interventions accumulate their effects, nor can we design principled strategies for therapies such as deep brain stimulation (DBS) or transcranial magnetic stimulation (TMS). **A data-driven, predictive, and testable description of plasticity rules at these timescales is therefore urgently needed**, for both mechanistic understanding and clinical optimization.

Existing models fall into two extremes. Classical biophysical rules, such as Hebbian learning (Hebb, 1949) and spike-timing–dependent plasticity (STDP) (Caporale & Dan, 2008; Dan & Poo, 2004), describe local synaptic changes on millisecond–minute scales. While interpretable, they are too narrow to capture circuit-level reorganization across long horizons (Zenke & Gerstner, 2017; Benna & Fusi, 2016; Frankland & Bontempi, 2005). Conversely, machine learning approaches allow connec-

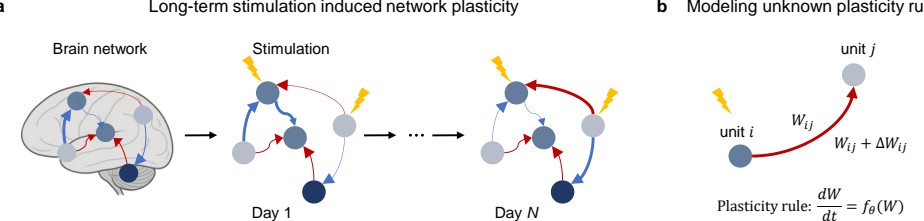

Figure 1: **Motivation**. Repeated stimulation induces slow evolution of network plasticity, with the underlying plasticity rules remaining unknown. The challenge of distinguishing slow plasticity from fast neural dynamics remains underdeveloped, with dynamical systems theory offering a potential framework to model these processes: $\frac{dW}{dt} = f_\theta(W)$.

tivity to vary across sessions (Pandarinath et al., 2018; Pellegrino et al., 2023; Vermani et al., 2025; Lu et al., 2025), but typically absorb slow change into high-dimensional parameters, entangling fast activity with slow adaptation. This entanglement impedes the identification of the underlying rule and limits generalization beyond the observed protocols. What is missing is **a general framework that uncovers slow plasticity rules and separates them from fast neural dynamics, as well as that reliably extrapolate to unseen stimulation protocols**.

We propose to treat long-term plasticity not as unstructured session-to-session variability but as a *latent dynamical law* driven by external protocols. Concretely, we introduce *stimulus-evoked evolution* (SE-Evolution) as plasticity rule: repeated stimulation drives a low-dimensional latent state $z_k$ that encodes plasticity embeddings. We formalize the SE-Evolution as

$$z_{k+1} = g_\theta(z_k, u_k),$$

where $u_k$ denotes the stimulation protocol and $g_\theta$ is the plasticity rule to be inferred. Once $g_\theta$ is identified, we can predict how connectivity and activity will evolve *under novel protocols*, enabling counterfactual design.

**STEER: a dual-timescale, identifiable formulation.** We develop *Stimulus-Evoked Evolution Recurrent dynamics* (STEER), a structured model that separates fast within-session activity from slow across-session adaptation. At the fast timescale, neural responses are generated by session-specific dynamics with structured recurrent connectivity capturing millisecond activity. At the slow timescale, latent coefficients $z_k$ evolve via a learnable recurrence $g_\theta$ that *is* the plasticity rule. STEER is trained to jointly optimize (i) within-session reconstruction, (ii) cross-session consistency, and (iii) structural regularizers that enforce motif interpretability. This yields *identifiable* separation between fast and slow processes and turns long-term plasticity from a confound into an explicit, testable object of inference. The main contributions of STEER are summarized as follows:

- **Conceptually novel:** We introduce the concept of *SE-Evolution*, framing long-term plasticity as a low-dimensional latent *dynamical law* inferred directly from longitudinal data.

- **Methodological advancement:** We develop **STEER**, a structured dual-timescale framework with priors that enable *identifiable* disentanglement of fast responses and slow network reconfiguration, yielding plasticity embeddings and motif-level readouts.

- **Predictive and out-of-distribution:** We show that the learned rule extrapolates to *unseen* stimulation protocols, predicting connectivity across (i) synthetic Lorenz systems, (ii) BCM-based neural models, (iii) task-learning through stimulation-induced plasticity and (iv) longitudinal Parkinson's DBS data, with accurate neural activity forecasts.

## 2 RELATED WORK

### 2.1 NEURAL SYSTEM IDENTIFICATION: FLEXIBLE SURROGATES, MISSING SLOW LAWS

Neural system identification grounded in dynamical systems theory (DST) provides powerful surrogates for uncovering latent flows in neural data (Durstewitz et al., 2023). RNN-based surrogates can

reconstruct nonlinear interactions and temporal dependencies from time series (Durstewitz et al., 2023; Luo et al., 2025), and low-rank designs improve sample efficiency and interpretability by constraining activity to low-dimensional manifolds (Mastrogiuseppe & Ostojic, 2018; Langdon et al., 2023; Valente et al., 2022; Pals et al., 2024). Methods such as SINDy further promote interpretable discovery of governing equations and primitives (attractors, limit cycles) (Brunton et al., 2016; Driscoll et al., 2024), while architectural/biophysical priors (e.g., neuromodulatory hypernetworks, E/I constraints) increase biological plausibility and identifiability (Song et al., 2016; Zhang et al., 2022; Achterberg et al., 2023; Costacurta et al., 2024).

**Limitation relative to our goal.** These works predominantly target *fast* neural dynamics within sessions. Even when session-to-session variability is allowed, the slow process is typically treated as unstructured parameter drift inside a high-dimensional model. As a result, (i) fast and slow processes are *entangled*, obscuring the plasticity *law*; (ii) generalization to *unseen stimulation protocols* is not a design objective; and (iii) there is no mechanism to make the slow rule *identifiable* rather than absorbed into flexible surrogates.

**Our motivation.** We elevate the slow process to a first-class target: a *stimulus-conditioned*, low-dimensional dynamical law that governs across-session reconfiguration. This requires (a) an explicit dual-timescale factorization to avoid entanglement, (b) structure/priors that restrict the hypothesis class of slow evolution, and (c) evaluation focused on unseen protocols.

## 2.2 OBTAINING PLASTICITY RULES: LOCAL/SHORT-TERM UPDATES VS. LONG-TERM RECONFIGURATION

A parallel line of work focuses on plasticity rules inference and derivation. Bredenberg et al. (2020) derived a task-dependent Hebbian plasticity rule to enable the learning of efficient, task-specific representations under resource and noise constraints. Kepple et al. (2022) explore how curriculum learning can be used to uncover the learning principles underlying both artificial neural networks and biological brains. Gradient-based parameterizations fit local synaptic update functions (Mehta et al., 2024). Meta-learning discovers self-organizing rules that stabilize sequence generation under synaptic turnover (Bell et al., 2024). These approaches illuminate *short-term* or *local* adjustments, often emphasizing immediate pre/post activity or reward signals.

**Limitation relative to long-term plasticity.** Long-term reconfiguration unfolds over days–weeks and recruits homeostatic control, synaptic scaling, and neuromodulatory pathways (Turrigiano & Nelson, 2004; Moulin et al., 2022; Marder, 2012; Liu et al., 2021). Prior methods typically: (i) focus on *microscopic* updates rather than the *mesoscopic* evolution of circuit motifs; (ii) do not explicitly condition the rule on external *stimulation protocols*, limiting counterfactual design; (iii) treat cumulative effects as aggregated parameter drift, which makes the governing slow rule *unidentifiable* and hampers extrapolation.

**Our motivation.** We recast the objective from "recover local updates" to "*learn a slow dynamical law* that maps stimulation histories to circuit-level coefficients." Doing so (i) aligns the modeling target with the experimental reality of longitudinal interventions, (ii) provides a natural handle for stimulation design, and (iii) opens the window to falsifiable rule-level predictions.

## 2.3 CLASSICAL BIOPHYSICAL RULES: INTERPRETABILITY WITHOUT HORIZON

Biophysical rules such as Hebbian learning and STDP (Hebb, 1949; Caporale & Dan, 2008; Dan & Poo, 2004) are mechanistically interpretable and operate on millisecond–minute scales. However, they struggle to capture circuit-level remodeling and memory consolidation across long horizons (Zenke & Gerstner, 2017; Benna & Fusi, 2016; Frankland & Bontempi, 2005), precisely where homeostasis and global neuromodulatory control become essential.

**Limitation relative to our goal.** Their temporal scope and locality make it difficult to predict multi-session consequences of repeated interventions, or to extrapolate across protocols that differ in schedule, intensity, or targeting.

**How STEER addresses these gaps:** (i) *From parameter learning to plasticity law.* We model across-session change as a low-dimensional, stimulus-evoked dynamical system, $z_{k+1} = g_\theta(z_k, u_k)$, rather than unstructured parameter drift. This centers the plasticity rule as the primary

object of inference. (ii) *Identifiable dual timescales.* A structured decomposition separates fast within-session responses from slow across-session evolution, augmented with causal/structural regularizers to promote identifiability of $f_\theta$. (iii) *Motif-level interpretability.* Reading out plasticity embeddings as recurrent motif scales provides mesoscopic, circuit-level summaries consistent with biological priors (e.g., E/I structure, neuromodulation).

## 3 METHOD

Inspired by neuroscience domain knowledge that high-dimensional neural recordings often admit low-dimensional latent structure (Valente et al., 2022), yet these structures themselves drift slowly across sessions during learning (Pellegrino et al., 2023) or long-term stimulation (Borra et al., 2024), we introduce STEER, a hierarchical model that parameterizes the fast recurrent dynamics within each session, the slowly varying, session-indexed connectivity governed by a learnable slow law. To this end, given $D$ ordered sessions with length-$T$ recordings $\{\mathbf{y}_{1:T}^k\}_{k=1}^D$, the goal of STEER is to use fast within-session dynamics to *identify* the slow, stimulus-evoked evolution that governs plasticity.

### 3.1 TIME-VARYING CONNECTIVITY VIA STRUCTURED LOW-RANK MOTIFS

We model $N$ neural recording units across $D$ sessions. Here, a session denotes a single contiguous block of population recording from the same preparation, over which we assume the recurrent connectivity is stationary, while it is allowed to change across sessions. For session $k \in \{1, \ldots, D\}$, $\mathbf{W}^k \in \mathbb{R}^{N \times N}$ is the recurrent connectivity used by the fast dynamics. Stacking $\{\mathbf{W}^k\}$ along the session axis yields a tensor $\mathcal{W} \in \mathbb{R}^{N \times N \times D}$, which we represent with a rank-$R$ CP decomposition,

$$\mathcal{W} = \sum_{r=1}^R \mathbf{a}_r \circ \mathbf{b}_r \circ \mathbf{c}_r, \quad \mathbf{a}_r, \mathbf{b}_r \in \mathbb{R}^N, \ \mathbf{c}_r \in \mathbb{R}^D. \tag{1}$$

Let $\mathbf{A} = [\mathbf{a}_1, \ldots, \mathbf{a}_R]$, $\mathbf{B} = [\mathbf{b}_1, \ldots, \mathbf{b}_R]$, and $\mathbf{C} = [\mathbf{c}_1, \ldots, \mathbf{c}_R] \in \mathbb{R}^{D \times R}$ with session coefficients $\mathbf{c}^k = \mathbf{C}_{k,:}^\top = (c_1^k, \ldots, c_R^k)^\top$. Then $\mathbf{W}^k = \sum_{r=1}^R c_r^k \mathbf{a}_r \mathbf{b}_r^\top, \quad k = 1, \ldots, D$.

**Identifiability structure.** To reduce CP scaling/sign indeterminacy, we (i) constrain $\|\mathbf{a}_r\|_2 = \|\mathbf{b}_r\|_2 = 1$ and absorb scales into $c_r^k$; (ii) penalize orthogonality with $\|\mathbf{A}^\top \mathbf{A} - \mathbf{I}\|_F^2 + \|\mathbf{B}^\top \mathbf{B} - \mathbf{I}\|_F^2$; and (iii) optionally impose Dale's principle (E/I structure) using a fixed sign mask $\mathbf{S} \in \{-1, 0, +1\}^{N \times N}$: $\mathbf{W}^k = \mathbf{S} \odot |\sum_{r=1}^R c_r^k \mathbf{a}_r \mathbf{b}_r^\top|$, which preserves prescribed signs.

### 3.2 FAST TIMESCALE: LOW-RANK RNN FOR WITHIN-SESSION DYNAMICS

To uncover the low-dimensional dynamics underlying the complex long-term neural recordings, we introduce a low-rank RNN as the core model to capture the shared latent state space. We encode the session-wise neural recordings $\mathbf{y}^k$ into the shared low-rank latent state space $\mathbf{h}^k$, where the latent state evolves as follows:

$$\tau \frac{\mathrm{d}}{\mathrm{d}t} \mathbf{h}^k(t) = -\mathbf{h}^k(t) + \left( \sum_{r=1}^R c_r^k \mathbf{a}_r \mathbf{b}_r^\top \right) \phi(\mathbf{h}^k(t)) + \mathbf{W}_{\mathrm{in}} \mathbf{u}^k(t). \tag{2}$$

Here, $\tau > 0$ is a time constant, $\phi(\cdot)$ is an element-wise nonlinearity, $\mathbf{u}^k(t)$ is the time-varying input for session $k$, and $\mathbf{W}_{\mathrm{in}}$ is the input matrix. A linear projector is used to decode the latent state $\mathbf{h}$ back to the observed space. During the implementation, we discretize equation 2 with Euler method.

### 3.3 SLOW TIMESCALE: STIMULUS-EVOKED EVOLUTION AS A LATENT LAW

**Session-level plasticity embedding.** We summarize each session by a plasticity embedding $\mathbf{z}^k \in \mathbb{R}^P$, obtained from data (and optionally inputs) via

$$\mathbf{z}^k = \mathrm{PlasticityEncoder}(\mathbf{y}_{1:T}^k, \mathbf{u}_{1:T}^k), \tag{3}$$

where the encoder can be permutation-aware across trials.

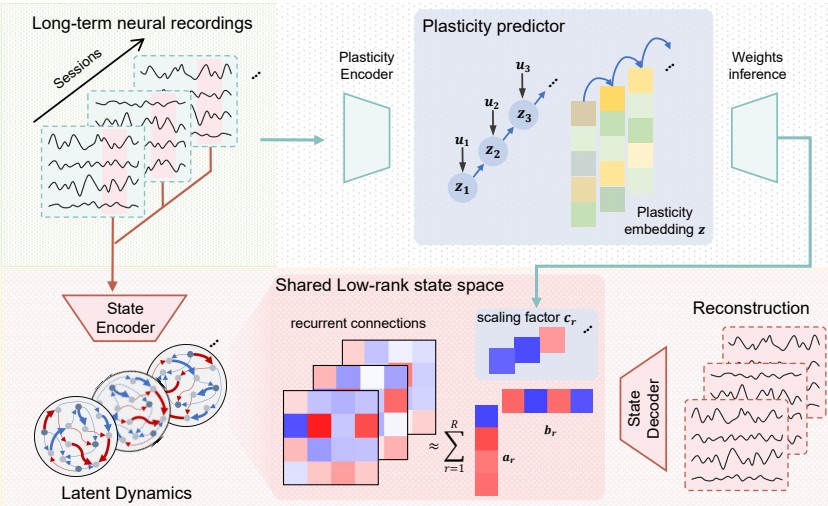

Figure 2: **STEER Framework**. A dual-timescale, identifiable formulation: within-session dynamics (fast) are generated by structured recurrent connectivity; across-session evolution (slow) follows a stimulus-conditioned latent law that we infer as the plasticity rule.

**Stimulus-conditioned slow law.** Across sessions, $\mathbf{z}^k$ follows a discrete-time residual evolution conditioned on a protocol summary $\bar{\mathbf{u}}^k = \mathrm{agg}(\mathbf{u}_{1:T}^k)$:

$$\mathbf{z}^{k+1} = g_\theta\big(\mathbf{z}^k, \bar{\mathbf{u}}^k\big) = \mathbf{z}^k + \tau_z\big(\mathbf{W}_z\,\phi(\mathbf{z}^k) - \mathbf{z}^k + \mathbf{B}_u\,\bar{\mathbf{u}}^k + \mathbf{b}_z\big). \tag{4}$$

**From embedding to motif scales.** Plasticity modulates motif strengths via a linear (or sparse) readout

$$\mathbf{c}^k = \mathbf{M}\mathbf{z}^k + \mathbf{b}_c, \qquad \mathbf{W}^k = \sum_{r=1}^{R} c_r^k\,\mathbf{a}_r\mathbf{b}_r^\top, \tag{5}$$

which is analogous to neuromodulatory gain control. Sparsity on $\mathbf{M}$ promotes interpretability (few embeddings per motif).

### 3.4 Training objective and optimization

We jointly learn fast and slow processes with a composite objective:

$$\mathcal{L} = \underbrace{\sum_{k=1}^{D}\sum_{t=1}^{T}\|\mathbf{y}_t^k - \hat{\mathbf{y}}_t^k\|_2^2}_{\mathcal{L}_{\text{fast}} \text{ (within-session multi-step prediction)}} + \lambda_{\text{slow}} \underbrace{\sum_{k=1}^{D-1}\big\|\mathbf{z}^{k+1} - g_\theta(\mathbf{z}^k, \bar{\mathbf{u}}^k)\big\|_2^2}_{\mathcal{L}_{\text{slow}} \text{ (across-session law consistency)}}$$

$$+ \lambda_{\text{smooth}} \underbrace{\sum_{k=1}^{D-1}\|\mathbf{c}^{k+1} - \mathbf{c}^k\|_2^2}_{\mathcal{L}_{\text{smooth}} \text{ (motif-scale smoothness)}}, \tag{6}$$

where $g_\theta$ is the right-hand side of equation 4. For $\mathcal{L}_{\text{fast}}$, we use $H$-step rollouts (teacher forcing at the first step, then fully autoregressive rollouts thereafter) to evaluate *multi-step* predictivity rather than one-step error, which better reflects dynamical fidelity over long horizons (Kramer et al., 2022; Brenner et al., 2022; 2024). The consistency term $\mathcal{L}_{\text{slow}}$ ensures the model captures the evolving plasticity embedding between sessions. The smoothness term $\mathcal{L}_{\text{smooth}}$ enforces gradual, continuous changes of motif scales between sessions. Since our framework has two timescales, we must choose how many steps into the future the model self-predicts the plasticity embedding before re-anchoring to data. This choice explicitly sets the amount of long-timescale teacher forcing and is needed as our model forecasts across sessions.

**Counterfactual protocols and OOD evaluation.** Given a novel stimulation protocol sequence $\{\tilde{\mathbf{u}}^k\}$, we compute $\tilde{\bar{\mathbf{u}}}^k$, iterate the slow law $\mathbf{z}^{k+1} = g_\theta(\mathbf{z}^k, \tilde{\bar{\mathbf{u}}}^k)$, obtain $\tilde{\mathbf{c}}^k$ via equation 5, assemble $\tilde{\mathbf{W}}^k$, and simulate equation 2 to predict neural outcomes. This separation explicitly tests whether the learned *rule* (not merely within-session responses) governs across-session reconfiguration and supports design of unseen protocols. To assess out-of-distribution behavior in a realistic setting, we adopt a forward-in-time evaluation protocol. Models are trained on the early portion of the long-term recording (first 60% of sessions) and tested on the later portion (last 40%). This temporal holdout mimics deployment where inference is made on future data and naturally captures distribution shifts over long timescales (e.g., shifts in stimulation/neural activity statistics).

## 4 EXPERIMENTS

To validate the STEER framework, we conducted a series of experiments on synthetic datasets and a real neural dataset. We generated synthetic data with ground truth using three standard benchmarks: the Lorenz dynamical system (Lorenz, 1963), the BCM plasticity model (Bienenstock et al., 1982), and a task-learning setting with adaptively optimized external stimulation (Borra et al., 2024). For the synthetic datasets, we sampled multivariate time series by varying the underlying ODE parameters. For the real data, we used longitudinal neural recordings from Parkinsonian rats receiving closed-loop DBS (Wang et al., 2024). We compared our method with other meta-learning based black-box dynamical models, a hierarchical PLRNN (Brenner et al., 2025) and a meta-dynamical state space model (MD-SSM) (Vermani et al., 2025) on synthetic datasets. We compared against a hierarchical PLRNN to test whether our low-rank inference can match a model that directly infers a full-rank weight matrix, because the original framework does not generalize to our setting, we use an adapted version (implementation details in Appendix A.3). We also compared against MD-SSM, which makes a similar low-rank assumption, to assess whether enforcing a session-invariant latent motif is important for recovering the plasticity rule, and whether our model can achieve both competitive fast-timescale accuracy and accurate slow-timescale forecasts (Appendix A.3).

### 4.1 LORENZ SYSTEMS WITH PARAMETER EVOLUTION

As a classical chaotic system, the Lorenz system (equation 7) exhibits complex nonlinear dynamics and is highly sensitive to initial conditions (butterfly effect), making it an ideal model for studying complex system dynamics. In this experiment, we aimed to model a plausible dynamic process that mirrors what is observed in biological neural networks, where synaptic connections can be strengthened or weakened. Therefore, we predefined plasticity rules that simulate the stimulus-evoked evolution of system parameters (Fig. 3a), which are expressed as equation 8.

$$
\begin{array}{lcl lcl}
\dot{x} & = & \sigma(y - x), & \sigma_m & = & \sigma_{\max} - \big(\sigma_{\max} - \sigma_{\text{init}}\big)e^{-m/40}, \\
\dot{y} & = & x(\rho - z) - y, \quad (7) & \rho_m & = & \rho_{\text{init}} - \dfrac{\ln(m + 10)}{10}, \\
\dot{z} & = & xy - \beta z. & \beta_m & = & \beta_{\text{init}} + \dfrac{m}{40}\ln\Big(\dfrac{m + 20}{20}\Big).
\end{array}
\qquad (8)
$$

where $\sigma_{\text{init}} = 10$, $\sigma_{\max} = 15$, $\rho_{\text{init}} = 48$, $\beta_{\text{init}} = 3.5$. The neural trajectories of the Lorenz systems were generated with a *fixed* initial state $[x(0), y(0), z(0)] = [1, 0, 20]$, ensuring that observed distributional shifts are solely attributable to the induced parameter changes, rather than initialization effects or added noise. The parameter ranges were selected to cover various dynamic regimes, each characterized by distinct attractor structures such as limit cycles and chaotic behaviors. For each set of parameters, time series were generated with a maximum length of $T_{max} = 10000$. In total, we generated Lorenz dynamics for 100 parameter settings $(\sigma_m, \rho_m, \beta_m)$ sampled along the schedule (with parameters held fixed within each system).

To optimize model performance, we first searched for the optimal parameters to determine the rank of the low-rank architecture. Since the three parameters showed no obvious correlation, we required a rank-3 low-rank model for an accurate approximation (Fig. 3b). This choice captures cross-system variations more effectively and improves predictive performance. The inferred motif scales are latent and therefore lack a direct one-to-one mapping to the ground-truth plasticity rule (Fig. 3c). We

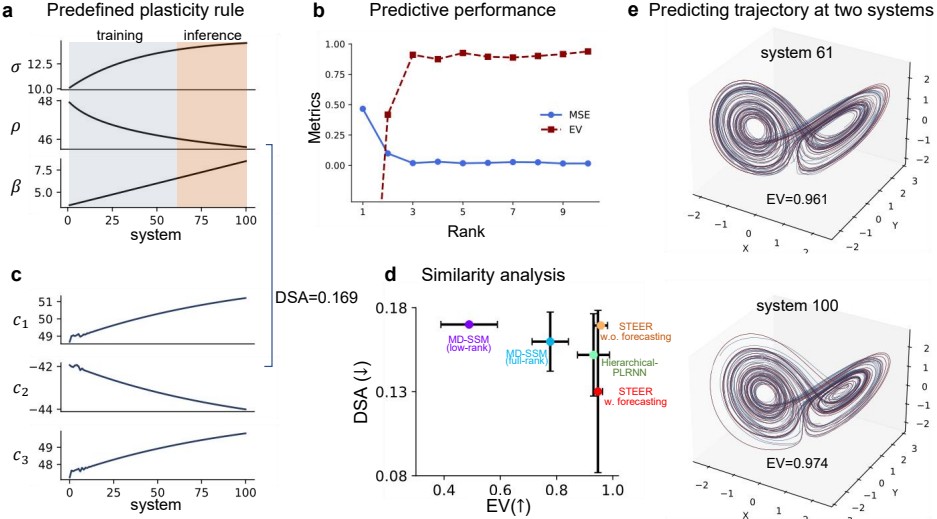

Figure 3: **STEER uncovers the parameter evolution in Lorenz system.** (a) Plasticity rules for system parameter evolution. (b) Rank selection for a low-rank model, with rank-3 providing optimal approximation. (c) Predicted implicit scaling factor and its similarity to the true plasticity rule (DSA = 0.169). (d) Similarity analysis among STEER and other baseline models (MD-SSM and hierarchical-PLRNN), with STEER having the lowest DSA and best EV ("w." and "w.o." denote with and without forecasting, respectively). (e) Prediction results for unseen data (systems 61 and 100) with high explained variance (EV = 0.961 and EV = 0.974).

quantified their correspondence using Dynamical Similarity Analysis (DSA) (Ostrow et al., 2023), yielding a score of 0.169. Meanwhile, we compare our framework with two baselines (MD-SSM and Hierarchical PLRNN) on both predictive performance (larger EV means better predictive capability) and dynamical similarity (lower DSA means better alignment). Fig. 3d indicates that STEER has better EV and lower DSA on session-wise prediction. To further validate the model's performance, we visualized the prediction results on unseen data (Fig. 3e). Both systems 61 and 100 exhibited high explained variance scores (EV = 0.961 and EV = 0.974, respectively), demonstrating the model's robust ability to predict unseen dynamics.

## 4.2 STIMULUS-EVOKED PLASTICITY IN BCM

To evaluate the performance of plasticity rule inference, we also simulated the dual-timescale dynamics via BCM rule as a controlled benchmark (Fig. 4a–b, Appendix Fig. 1). In this setting, the plasticity mechanism and stimulus-evoked modular organization are known, enabling direct assessment of the inferred network reconfiguration across sessions. We modeled the stimulus-evoked modifications of the recurrent weights across sessions through BCM plasticity rule (Fig. 4a). For session $k + 1$,

$$\mathbf{W}_{hh}^{k+1} = \mathrm{Proj}_{\mathrm{E/I}}\big(\mathbf{W}_{hh}^{k} + \eta_W \left(\bar{\mathbf{h}}^{k} \odot (\bar{\mathbf{h}}^{k} - \theta^{k})\right) (\bar{\mathbf{h}}^{k})^{\top}\big), \tag{9}$$

where $\odot$ denotes element-wise multiplication. $\mathbf{h}^{k}$ is neuronal activities and $\theta^{k}$ is a neuron-specific sliding threshold that sets the potentiation–depression boundary, and itself adapts slowly across sessions (Fig. 4b). We preserved the E/I partition in $\mathbf{W}_{hh}$ by fixing outgoing-weight signs ($E \geq 0, I \leq 0$). For session $k$, the neuronal dynamics were governed by a leaky-integrator recurrent network with fixed $\mathbf{W}_{hh}^{k}$ and $\theta^{k}$ (details in Appendix A.1).

We first compared STEER with MD-SSM variants. All models achieved similarly high within-session predictive accuracy across sessions (EV > 0.9; Fig. 4d), with no significant short-term differences once capacity is matched (t=0.4221, p>0.05). In contrast, STEER more accurately recovered stimulus-evoked cross-session changes in recurrent connectivity: the inferred $\|\mathbf{W}\|_2$ closely followed the ground truth, whereas MD-SSM systematically deviated (Fig. 4c). This suggests that treating each session as effectively independent, or only meta-learning trial-level variability, discards

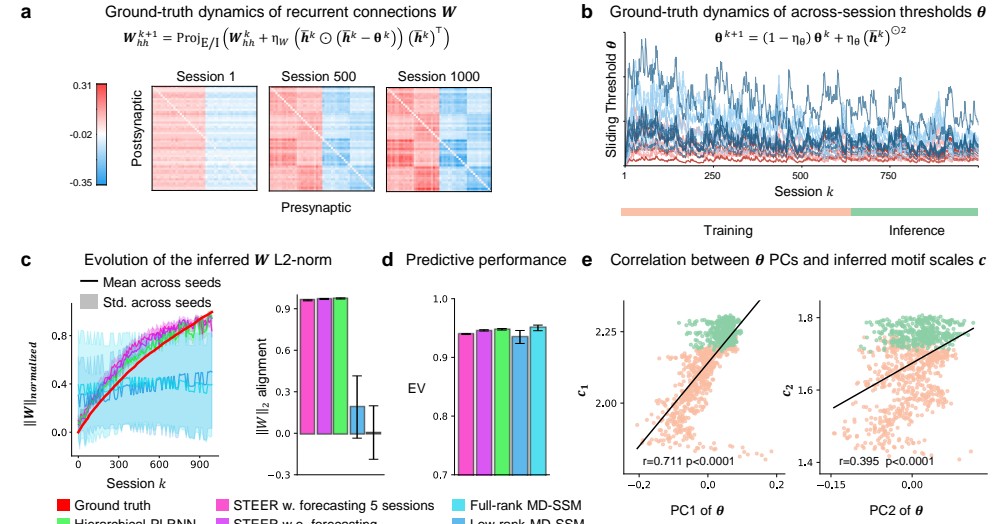

Figure 4: **STEER captures stimulus-evoked plasticity in BCM.** (a & b) We generate data via a BCM-driven across-session simulation, subsequently fitting models with a 600/400 train–test split. (c) Inferred $\|\mathbf{W}\|_2$ evolution over sessions alignment with the ground truth. Left: Normalized L2-norm of inferred vs. true $\mathbf{W}$ over sessions (mean ± std. across random seeds). Right: $\|\mathbf{W}\|_2$ alignment (correlation between the session-by-session trajectories of the inferred and ground-truth $\|\mathbf{W}\|_2$), comparing STEER with and without forecasting to adapted hierarchical-PLRNN, full- and low-rank MD-SSM baselines. (d) Within-session predictive performance (EV) on all sessions (mean ± std. across random seeds), showing that STEER achieves competitive fast-timescale prediction accuracy. (e) STEER inferred motif scales $\mathbf{c}$ significantly correlates with the ground-truth sliding thresholds $\theta$ principal components.

inter-session structure that is essential for identifying slow network reconfiguration. By explicitly modeling coupled fast and slow timescales through a shared latent motif, STEER preserves this information and yields a more faithful reconstruction of plasticity. Consistently, correlations between $\mathbf{c}$ and the leading PCs of the ground-truth thresholds $\theta$ (Fig. 4e) show that $\mathbf{c}$ captures the same low-dimensional homeostatic manifold. Thus, $\mathbf{c}$ serves as an order parameter for stimulus-history–dependent plasticity, supporting timescale disentanglement.

To further test slow-timescale identifiability, we evaluated a STEER variant that forecasts slow parameters 5 sessions ahead, alongside its non-forecast counterpart and the full-rank hierarchical PLRNN baselines. Even in this harder setting, STEER preserved strong fast-timescale accuracy and closely tracked session-to-session changes in $\mathbf{W}$. Moreover, the non-forecast STEER variant did not differ significantly in either slow- or fast-timescale performance from the full-rank hierarchical model. However, the inferred recurrent weights (Appendix Fig. 2) showed that the hierarchical PLRNN baseline yielded much weaker modular structure, whereas the low-rank models (STEER and MD-SSM) recovered modules that closely match the ground-truth organization. This agreement highlights the advantage of a low-rank structure for recovering stimulus-dependent circuit motifs shaping population dynamics.

### 4.3 TASK LEARNING THROUGH STIMULATION-INDUCED PLASTICITY

We construct a synthetic benchmark (Borra et al., 2024) to test whether STEER can recover *stimulation-evoked* reconfiguration of recurrent connectivity from population dynamics **when the external drive is itself optimized in closed loop** (see Appendix A.1). This setting contrasts with our BCM dataset, where synaptic updates follow a fixed plasticity rule under **prespecified** inputs: here, changes in $\mathbf{W}(t)$ arise from adaptive control, and the resulting neural trajectories are shaped by a learned stimulus policy rather than by a hand-designed input ensemble. A recurrent rate network with time-varying connectivity $\mathbf{W}(t)$ is driven by control inputs $\mathbf{u}(t)$, which are adaptively optimized across learning cycles $k = 0, \ldots, K$ to steer $\mathbf{W}$ toward a ring-attractor target $\mathbf{W}^{\text{target}}$.

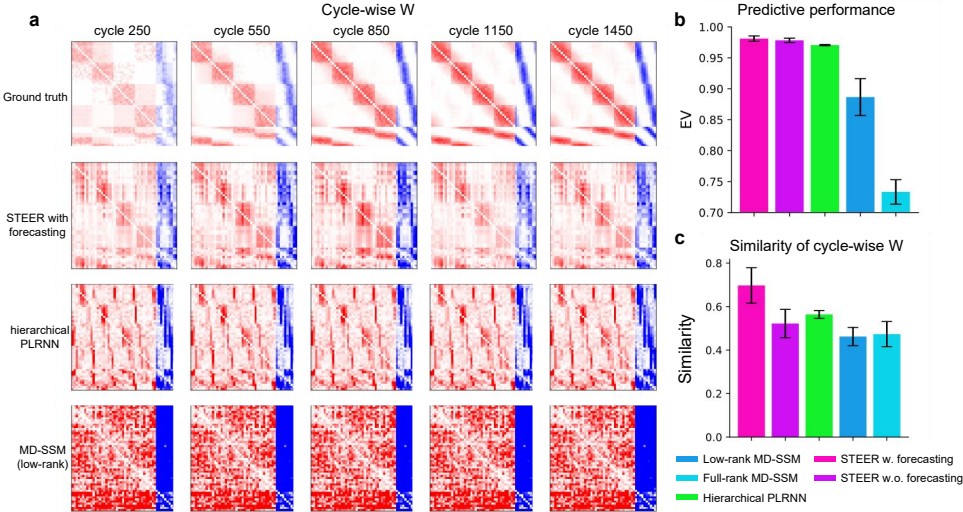

Figure 5: **STEER captures stimulus-evoked plasticity in cycle-wise task learning.** (a) Cycle-wise weight dynamics during task learning and inference. (b) Predictive performance across various methods. (c) Similarity between cycle-wise weights in model inference and ground truth. STEER with forecasting demonstrates exceptional performance in prediction accuracy and weight similarity.

We first searched for the optimal rank of the low-rank structure of STEER. A rank-7 STEER model was necessary to achieve an accurate approximation and high similarity (Appendix Fig. 11). The STEER model effectively captures stimulus-evoked plasticity throughout cycle-wise task learning. As shown in Fig. 5a, the cycle-wise weight dynamics during task learning and inference highlight that STEER is capable of inferring module-like weights. In Fig. 5b and c, the predictive performance in fast-timescale neural dynamics and slow-timescale similarity between cycle-wise weights in the model inference and the ground truth emphasize STEER's exceptional performance in prediction accuracy and replicating the expected weight dynamics.

## 4.4    PD-DBS Longitudinal Neural Dataset

We further validated the proposed framework in the publicly available longitudinal electrophysiology and behavior dataset of Parkinson's disease (PD) rats undergoing closed-loop deep brain stimulation (Wang et al., 2024). The original dataset study validated the efficacy of DBS treatment in alleviating the symptoms of PD. This dataset is well-suited to our study of inferring brain plasticity rules based on long-term neural activity by quantifying dynamical and plastic changes across different cohorts. The dataset comprises three cohorts (Sham, PD without treatment, and PD with DBS) and was recorded from week 2 to week 5 (four weekly time points, Fig. 6a). Signals were collected from layer V of primary motor cortex, providing raw wideband along with derived spikes and local field potentials (LFPs), accompanied by open-field behavior videos and quantitative measures (see Appendix A.1). The PD-DBS cohort received closed-loop STN-DBS on days 29 to 33 with simultaneous neural recording, enabling comparisons within subjects before, during and after stimulation (Fig. 6a).

After modeling, we revealed that a strong alignment between the inferred motif scales $\mathbf{c}$ and functional connectivity (FC), indicating a high level of consistency between plasticity factors and FC (Fig. 6c, see Appendix A.4). Further, we assessed the cosine similarity between $\Delta\mathbf{c}$ and $\Delta FC$, showing that all groups had a similar cosine similarity between $\Delta\mathbf{c}$ and $\Delta FC$ across weeks, supporting that $\Delta\mathbf{c}$ is closely aligned with the intrinsic functional connectivity (Fig. 6d, see Appendix A.4). We also present the changes in the magnitude of $\Delta\mathbf{c}$ ($\Delta\mathbf{c}_{\mathrm{mag}}$) across different groups (PD, PD-DBS, Sham) over the DBS intervention interval ($w_4 \to w_5$), revealing that the PD-DBS group exhibited significantly higher $\Delta\mathbf{c}_{\mathrm{mag}}$ values compared to the PD and Sham groups (Fig. 6e). This suggests that DBS stimulation effectively promotes the improvement of neural activity in PD rats. Overall, these

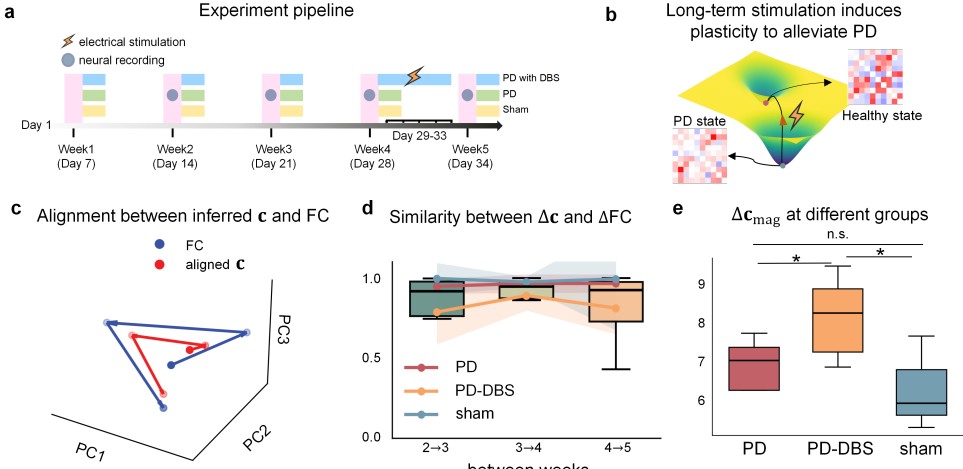

Figure 6: **STEER interprets long-term DBS effect in PD rats.** (a) Longitudinal experimental pipeline: repeated recordings across weeks 2–5; PD–DBS receives closed-loop STN–DBS on days 29–33. (b) Long-term stimulation induces functional connectivity plasticity to alleviate the symptoms of PD. (c) Alignment between FC trajectories (blue, PCA space) and STEER motif scales $\mathbf{c}^k$ (red) across weeks. (d) Week-to-week cosine similarity between $\Delta\mathbf{c}$ and $\Delta FC$. (e) Magnitude of plasticity increment ($\Delta\mathbf{c}_{\mathrm{mag}} = \|\Delta\mathbf{c}\|_2$). DBS shows larger changes, consistent with a stimulation-evoked slow update.

findings suggest that DBS is associated with a distinct plasticity-related neural signature, consistent with modulation of neural network plasticity.

## 5 DISCUSSION AND CONCLUSION

In this study, we presented the STEER framework to learn *plasticity rules* from neural recordings under long-term stimulation. STEER formalizes long-term plasticity as a stimulus-conditioned latent dynamical system, $z_{k+1} = g_\theta(z_k, \bar{\mathbf{u}}_k)$, and enforces an identifiable separation between fast within-session responses and slow network reconfiguration.

**Methodological impact.** By leveraging the dual-timescale nature of neural dynamics, STEER effectively separates fast, within-session neural responses from slower, session-wise plasticity changes. Across synthetic Lorenz dynamics, BCM-driven networks, task-learning benchmark and longitudinal PD–DBS recordings, this formulation consistently turns "unstructured variability" into *protocol-indexed trajectories* in a low-dimensional plasticity space, yielding interpretable motif-level readouts and out-of-distribution predictions under unseen stimulation schedules.

**Limitations and opportunities.** First, our presented behavioral analyses are limited. Linking $\mathbf{c}$-space trajectories to subject-level outcomes (e.g., motor scores) via mixed-effects models and prospective perturbations will be crucial. Second, richer biological structure (e.g., cell-type–specific gains, homeostatic set points, neuromodulator-dependent gating) could be embedded directly into $g_\theta$ and the $\mathbf{c} \to W$ readout. Third, safety-aware design requires constraints on reachable sets of $\mathbf{z}$ and $\mathbf{c}$ (e.g., contractive regions and Lyapunov margins). Finally, broader validation across modalities (optogenetics, TES, TMS), species, and disease states will test the portability of the learned rules.

**Outlook.** By making the slow rule explicit, STEER turns long-term plasticity from a hidden confound into a falsifiable object. We anticipate two immediate payoffs: (i) *mechanistic inference*, where motif-level embeddings act as mesoscopic biomarkers of neuromodulatory control; and (ii) *closed-loop design*, where protocol parameters are optimized in the space of laws rather than trajectories. We believe this shift, from parameter drift to learned dynamical law, offers a principled foundation for next-generation neurostimulation that is both interpretable and adaptive.

ACKNOWLEDGMENTS

This work was supported by Brain Science and Brain-like Intelligence Technology-National Science and Technology Major Project (2021ZD0200500), the National Natural Science Foundation of China (62472206, 3254100307), National Key R&D Program of China (2025YFC3410000), Shenzhen Science and Technology Innovation Committee (RCYX20231211090405003, JCYJ20220818100213029, JCYJ20240813141105007), Guangdong Basic and Applied Basic Research Foundation (2025A1515011645, 2026B1515020099), Guangdong S&T Program (Grant No. 2026B0101110003), Shanghai Municipal Special Program for Basic Research on General AI Foundation Models (2025SHZDZX026D05), and the open research fund of the Guangdong Provincial Key Laboratory of Mathematical and Neural Dynamical Systems, the Center for Computational Science and Engineering at Southern University of Science and Technology, Shenzhen Key Laboratory of Smart Healthcare Engineering, SUSTech Undergraduate Innovation and Entrepreneurship Training Program (2024X08).

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

# A APPENDIX

## A.1 DATASET DETAILS

**Lorenz system.** The famous 3D Lorenz attractor (Lorenz, 1963) is widely used as a benchmark in data-driven system research. Its governing equations are

$$
\begin{aligned}
\dot{x} &= \sigma\,(y - x), \\
\dot{y} &= x(\rho - z) - y, \\
\dot{z} &= xy - \beta z.
\end{aligned}
\tag{1}
$$

**BCM plasticity rule.** The BCM rule employs a sliding threshold mechanism to dynamically adjust the plasticity gate. It achieves homeostatic regulation based on past activity levels. The update of the threshold $\theta$ is specified by:

$$
\theta^{k+1} = (1 - \eta_\theta)\,\theta^k + \eta_\theta\,(\bar{\mathbf{h}}^k)^{\odot 2},
\tag{2}
$$

where $x^{\odot 2}$ denotes element-wise squaring, $\eta_\theta \in (0,1)$ is the smoothing rate and $\bar{\mathbf{h}}^k$ is the mean activity vector in session $k$:

$$
\bar{\mathbf{h}}^k := \frac{1}{T} \sum_{t=0}^{T-1} \mathbf{h}_t^k,
\tag{3}
$$

where $T$ denotes the session length. The neuron activity in each session $k$ is defined by

$$
\mathbf{h}_{t+1}^k = \mathbf{h}_t^k + \frac{\delta}{\tau_h}\Big( -\mathbf{h}_t^k + \mathbf{W}_{hh}^k \tanh(\mathbf{h}_t^k) + \mathbf{b} + \mathbf{W}_{in}\,\mathbf{u}_t^k \Big)
\tag{4}
$$

where $\mathbf{u}(t)$ is a two-channel, piecewise-constant one-hot stimulus alternating across channels (Appendix Fig. 1). We discretized the dynamics with a forward-Euler step of size $\delta$ (set to $\delta = 1$ in practice) and update the state at each step. This threshold implements a homeostatic control mechanism (Abbott & Dayan, 2001). The dynamics can be interpreted as follows: when the mean activity $\bar{\mathbf{h}}^{\cdot k}$ is persistently elevated, $\theta$ increases, thereby raising the plasticity threshold and counteracting runaway potentiation. Conversely, reduced activity drives $\theta$ downward, permitting synaptic strengthening. This negative feedback loop ensures stability while maintaining sensitivity to changes in input statistics.

Together with the weight update rule in equation 9, the sliding threshold allows the model to capture both activity-dependent potentiation/depression and homeostatic regulation across sessions. In our simulations, this mechanism prevents weights from saturating and enables the emergence of stable modular connectivity patterns under repeated stimulation. Because the BCM rule is Hebbian in nature, coactivation of units (Appendix Fig. 1) systematically strengthens their mutual connections (Appendix Fig. 2), leading to a gradual increase in the norm of the recurrent weight matrix over sessions.

**Task Learning through Stimulation-Induced Plasticity.** To construct a dataset in which recurrent connectivity evolves under biologically motivated plasticity, we adopt the rate-based network and plasticity model of (Borra et al., 2024). The network contains $N$ neurons with firing rates $\mathbf{r}(t) = (r_1(t), \ldots, r_N(t))$ and a time-dependent recurrent connectivity matrix $\mathbf{W}(t) = (W_{ij}(t))_{i,j=1}^N$. Neural activity obeys the standard rate dynamics

$$
\tau_n \frac{dr_i}{dt}(t) = -r_i(t) + \phi\left( \sum_j W_{ij}(t)\,r_j(t) + u_i(t) \right),
\tag{5}
$$

where $\tau_n$ is the membrane time constant, $u_i(t)$ is an externally applied control input on neuron $i$, and $\phi(\cdot)$ is a sigmoidal input–output transfer function saturating at a maximal firing rate $r_{\max}$. Similarly, the control signals $u_i(t)$ are chosen in the same form as in the original work, and are re-optimized across learning cycles to steer connectivity towards a predefined target structure.

Synaptic plasticity is modeled as a Hebbian covariance rule with homeostatic feedback and soft bounds on synaptic amplitudes. The dynamics of each synaptic weight $W_{ij}(t)$ is given by

$$
\begin{aligned}
\tau_s \frac{dW_{ij}}{dt}(t) = {} & \eta(\alpha_j)\,\big(r_i(t) - \theta(\alpha_j)\big)\,r_j(t) \; - \; \beta_1\,|W_{ij}(t)|\big(r_i^2(t) - \theta_0(\alpha_j)^2\big) \\
& - \beta_2\,\mathrm{sgn}\big(W_{ij}(t)\big)\,h\big(|W_{ij}(t)| - \bar{W}\big),
\end{aligned}
\tag{6}
$$

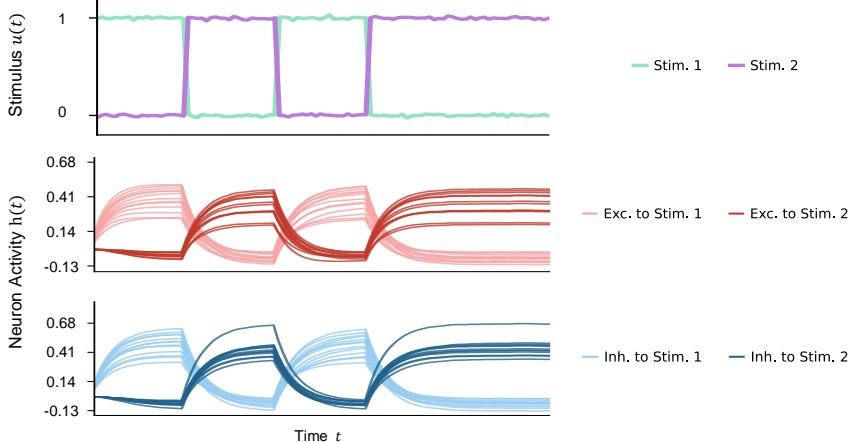

Appendix Fig. 1: **Stimulus selectivity in simulated neuronal populations.** Simulated neuronal populations show clear stimulus selectivity, with neurons responding preferentially to specific stimuli and exhibiting structured co-activation patterns.

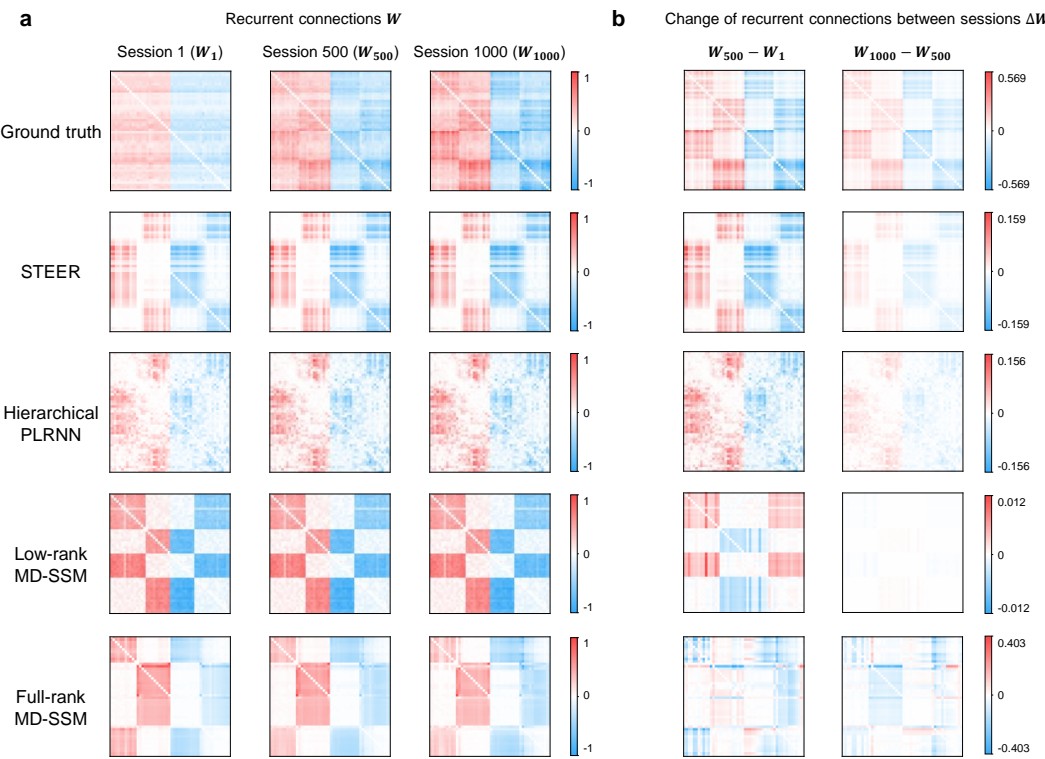

Appendix Fig. 2: Recurrent connection inference. (a) Recurrent weight matrices $\mathbf{W}$ at three time points (sessions 1, 500, and 1000) for the ground truth, STEER, full-rank MD-SSM, and low-rank MD-SSM. (b) Corresponding changes in connectivity between sessions ($\mathbf{\Delta W}$).

where $\alpha_j \in \{E, I\}$ denotes the type (excitatory/inhibitory) of presynaptic neuron $j$, $\eta(\alpha_j)$ is the learning rate for synapses from that neuron type, and $\theta(\alpha_j)$ and $\theta_0(\alpha_j)$ are activity thresholds that stabilize the postsynaptic firing rate. The function

$$h(x) = \begin{cases} x^2, & x \geq 0, \\ 0, & x < 0, \end{cases} \tag{7}$$

implements a soft clipping of synaptic strengths at magnitude $\bar{W}$, while $\beta_1$ and $\beta_2$ control the strengths of the two homeostatic terms. We work in the slow-plasticity regime $\tau_s \gg \tau_n$, so that firing rates are (quasi-)stationary on the timescale over which $\mathbf{W}(t)$ changes.

Following (Borra et al., 2024), we consider a structural target in which the excitatory and inhibitory neurons are arranged on concentric rings and the target connectivity $\mathbf{W}^{\text{target}}$ supports a continuous ring attractor. Training aims to reshape the initial random connectivity $\mathbf{W}^{(0)}$ towards $\mathbf{W}^{\text{target}}$ by optimizing the control inputs $\{u_i(t)\}$ over a sequence of learning cycles $k = 0, 1, \ldots, K$. In each cycle, the cost

$$L_{\text{task}}(\mathbf{W}) = \sum_{i,j} w_{\alpha_i, \alpha_j} \left( W_{ij} - W_{ij}^{\text{target}} \right)^2 \tag{8}$$

penalizes deviations from the ring-attractor connectivity, with type-dependent weights $w_{\alpha_i, \alpha_j}$ that balance contributions from all four connection classes (E→E, E→I, I→E, I→I). Given an estimate of the current connectivity $\mathbf{W}^{(k)}$, we numerically optimize the control stimulation $u_i^{*,k}(t)$ over a fixed control period $t \in [0, T_{\text{ctrl}}]$ to approximately minimize $L_{\text{task}}(\mathbf{W}^{(k+1)})$ under the plasticity dynamics in equation 6. Applying $u^{*,k}$ generates a trajectory of firing rates $\mathbf{r}^{(k)}(t)$ and an updated connectivity matrix $\mathbf{W}^{(k+1)}$.

Following (Borra et al., 2024), we model the effect of a fixed control stimulation $\mathbf{u}(t)$ applied for a duration $T_{\text{ctrl}}$ starting from connectivity $\mathbf{W}^{(k)}$ as a plasticity-induced update $\Delta\mathbf{W}(\mathbf{u}, T_{\text{ctrl}}, \mathbf{W}^{(k)})$, obtained by integrating the rate and plasticity dynamics in equations 5 and 6. The optimal control for learning cycle $k$ is then defined as

$$\mathbf{u}^{*,k} = \arg\min_{\mathbf{u}} L_{\text{task}}\left( \mathbf{W}^{(k)} + \Delta\mathbf{W}(\mathbf{u}, T_{\text{ctrl}}, \mathbf{W}^{(k)}) \right), \tag{9}$$

which we solve numerically by gradient descent in the space of control signals, while keeping the Hebbian plasticity rule fixed.

To generate our dataset, we simulate the coupled fast–slow dynamics starting from an initially random connectivity $\mathbf{W}^{(0)}$ while iteratively optimizing the stimulation controls across learning cycles. For each cycle $k$, we record the evoked activity trajectory $\{\mathbf{r}^{(k)}(t)\}_{t=0}^{T_{\text{ctrl}}}$ under the optimized protocol $u^{*,k}$, together with the pre- and post-cycle connectivity matrices $\mathbf{W}^{(k)}$ and $\mathbf{W}^{(k+1)}$ as ground truths across sessions, which progressively approach the ring-attractor structure (Fig. 5). In our experiments, we use $\{\mathbf{r}^{(k)}(t)\}$ and the optimized $u^{*,k}$ as input to STEER, directly testing whether stimulation-driven connectivity reorganization can be inferred from neural activity alone.

**PD-DBS Rat Dataset.** We use the public dataset *"A longitudinal electrophysiological and behavior dataset for PD rat in response to deep brain stimulation"* (Wang et al., 2024), released on the GIN (G-Node) repository at `https://doi.org/10.12751/g-node.lzvqb5`. The experimental protocol spanned five weeks and included three groups of animals: *sham*, *PD*, and *PD-DBS*.

- **Sham group.** Rats underwent the same surgical implantation of stimulation and recording electrodes as other groups, but instead of the neurotoxin, they received a vehicle injection (ascorbic acid in saline) into the medial forebrain bundle (MFB; AP $\sim$ –4.4 mm, ML $\sim$ –1.2 mm, DV $\sim$ –7.8 mm). Recording electrodes were implanted only in the left primary motor cortex M1 (AP +2.5 mm, ML +3.0 mm, DV –1.6 mm), and stimulation electrodes in left STN (AP –3.6 mm, ML –2.6 mm, DV –7.6 mm). Sham rats never developed hemi-parkinsonism and did not receive DBS, serving as surgical and handling controls.

- **PD group.** Rats received unilateral 6-hydroxydopamine (6-OHDA; 4 $\mu$l, 5 $\mu$g/$\mu$l) injection into the left MFB to induce hemi-Parkinsonism, confirmed on Day 6 with apomorphine-rotation screening (net >210 contralateral turns/30 min). Both stimulation and bilateral

M1 recording electrodes were implanted as in the DBS group, but no DBS was applied. These rats allowed assessment of electrophysiological and behavioral alterations due to PD pathology without stimulation.

- **PD-DBS group.** Rats received the same unilateral 6-OHDA lesion and electrode implantation as the PD group. In addition, during Days 29–33 (Weeks 4–5), they underwent closed-loop STN-DBS (130 Hz, 90 $\mu$s, current individualized by motor threshold) with simultaneous recording. The closed-loop controller used spectral features of M1 LFPs (2–50 Hz) to trigger stimulation. This group provided the longitudinal neural and behavioral data under chronic adaptive DBS.

Across all groups, electrodes were secured with dental acrylic, and placement was histologically verified post-mortem. Longitudinal data were organized into three daily recording episodes (morning/afternoon/night) with synchronized open-field behavior. In our study, for consistent temporal comparison of neural dynamics and plasticity across cohorts, we selected only the morning episodes (episode 1) from Weeks 2–5. For the PD-DBS group, all electrophysiology during the DBS intervention window (Days 29–33) was excluded. Within each selected episode, recordings were further segmented by behavioral state (sleep/wake/walk), and we restricted our analyses to the walk segments, which most clearly differentiate PD and non-PD animals. Otherwise, we adhered to the dataset's native channel mapping and indexing conventions.

## A.2   SCALING EFFECT FOR NETWORK CONNECTIONS IN STEER

To simulate the phenomenon of neural plasticity induced by long-term stimulation, we infer the motif scales $\mathbf{c}^k$ from neural signals, akin to neuromodulatory signals, which then influences the network's connectivity $\mathbf{W}^k = \sum_{r=1}^{R} c_r^k \mathbf{a}_r \mathbf{b}_r^T$. The motif scales inferred from plasticity embedding act as adaptive regulators of synaptic strength, where they either strengthen or weaken specific recurrent connections within the network. This dynamic modulation allows for the systematic reinforcement or weakening of neural pathways, which in turn drives the plasticity effects in the neural system. Through this mechanism, the connectivity weights evolve in response to the slow, long-term changes in the neural embedding, reflecting the neural system's capacity for learning, adaptation, and memory formation. The evolving plasticity factors thus enable a flexible and context-sensitive adaptation of the network's structure over time, facilitating the emergence of complex neural behaviors in response to environmental or internal stimuli.

## A.3   BASELINE IMPLEMENTATION DETAILS

**Hierarchical PLRNN.**   The original hierarchical PLRNN learns session-specific parameters directly rather than inferring session embeddings from the observed signals, and therefore does not generalize to unseen sessions in our setting. To make it comparable to STEER, we adapted its framework so that session embeddings are inferred from the observations in the same way as in our model, and these embeddings are then used to generate session-specific **full-rank** weights by projection.

**MD-SSM.**   Structurally, MD-SSM augments a session-specific low-rank transition with a shared full-rank recurrent matrix $W_{hh} \in \mathbb{R}^{N \times N}$,

$$W_{Full-rank\ MD-SSM}^{(s)} = U^{(s)} V^{(s)\top} + W_{hh}, \tag{10}$$

resulting in at least $N^2$ additional parameters compared to STEER, which uses session embeddings only to modulate coefficients of a shared low-rank tensor. To control for learnable parameter count, we also evaluated a purely low-rank variant obtained by removing $W_{hh}$, while keeping all other modeling choices identical.

$$W_{Low-rank\ MD-SSM}^{(s)} = U^{(s)} V^{(s)\top}, \tag{11}$$

This variant isolates the effect of the shared full-rank recurrent dynamics from the meta-dynamical, session-specific components.

## A.4   PD-RAT ANALYSIS

**Alignment between model plasticity and FC.** After modeling, we observed a strong *directional alignment* between the inferred plasticity embedding $\mathbf{c}$ and empirical functional connectivity (FC)

across weeks (Fig. 6c). Concretely, we embedded week-wise FC (Fisher-$z$ upper triangles) and $\mathbf{c}$ (subject-wise $z$-scored across weeks) into 3D using PCA to obtain trajectories $\{\mathbf{f}_t\}_{t=1}^T$ and $\{\mathbf{c}_t\}_{t=1}^T$ for each subject. Because the PCA for FC and $\mathbf{c}$ was performed separately, the two 3D coordinate systems are only defined up to an arbitrary rotation/reflection and global scale, making their trajectories not directly comparable. To remove these nuisance degrees of freedom and focus on trajectory *shape*, we aligned the plasticity trajectory to the FC trajectory using a full Procrustes similarity transform.

Let $F \in \mathbb{R}^{T \times 3}$ and $C \in \mathbb{R}^{T \times 3}$ denote the stacked PCA coordinates with rows $F_t^\top = \mathbf{f}_t^\top$ and $C_t^\top = \mathbf{c}_t^\top$. For each subject we solved the standard Procrustes problem

$$(\mu_f, \mu_c, R, s) \;=\; \arg \min_{\mu_f, \mu_c \in \mathbb{R}^3, \; R^\top R = I, \; s > 0} \left\| F - \left( \mathbf{1}\mu_f^\top + s\,(C - \mathbf{1}\mu_c^\top)R \right) \right\|_F^2, \qquad (12)$$

$\mu_f$ and $\mu_c$ are the row means of $F$ and $C$, respectively; $R$ is obtained from the SVD of $(C - \mathbf{1}\mu_c^\top)^\top (F - \mathbf{1}\mu_f^\top)$; and $s = \|F - \mathbf{1}\mu_f^\top\|_F / \|C - \mathbf{1}\mu_c^\top\|_F$. The aligned plasticity coordinates are then defined as:

$$\tilde{\mathbf{c}}_t \;=\; s\,R\big(\mathbf{c}_t - \mu_c\big) + \mu_f. \qquad (13)$$

Directional consistency was quantified by the absolute cosine similarity between consecutive displacement vectors of the FC and aligned plasticity trajectories,

$$\rho_t \;=\; \left| \frac{(\Delta\mathbf{f}_t)^\top (\Delta\tilde{\mathbf{c}}_t)}{\|\Delta\mathbf{f}_t\|_2 \, \|\Delta\tilde{\mathbf{c}}_t\|_2} \right| \in [0, 1], \qquad \Delta\mathbf{f}_t = \mathbf{f}_{t+1} - \mathbf{f}_t, \; \Delta\mathbf{c}_t = \tilde{\mathbf{c}}_{t+1} - \tilde{\mathbf{c}}_t. \qquad (14)$$

We take the absolute value because Procrustes alignment is defined up to reflection, which can flip the sign of displacement vectors without changing trajectory shape.

**Change magnitudes and their relation.** To relate plasticity changes to FC changes, we computed week-to-week change vectors $\Delta\mathbf{f}_t = \mathbf{f}_{t+1} - \mathbf{f}_t$ and $\Delta\mathbf{c}_t = \tilde{\mathbf{c}}_{t+1} - \tilde{\mathbf{c}}_t$, and summarized their magnitudes by $\ell_2$ norms, $\Delta\mathrm{FC}_{\mathrm{mag}}(t) = \|\Delta\mathbf{f}_t\|_2$ and $\Delta\mathbf{c}_{\mathrm{mag}}(t) = \|\Delta\mathbf{c}_t\|_2$. Across weeks, all groups showed similar trends in the relation between $\Delta\mathbf{c}_{\mathrm{mag}}$ and $\Delta\mathrm{FC}_{\mathrm{mag}}$ (Fig. 6d), indicating that the learned plasticity closely tracks intrinsic FC dynamics.

**Group comparison (DBS intervention interval).** We compared the week-to-week plasticity change magnitude $\Delta\mathbf{c}_{\mathrm{mag}}$ over the *DBS intervention interval* ($w_4 \rightarrow w_5$), corresponding to the period during which DBS was applied. The PD-DBS group showed significantly larger $\Delta\mathbf{c}_{\mathrm{mag}}$ than both the PD and Sham groups (two-sample $t$-tests, $p < 0.05$; Fig. 6e), consistent with stronger reorganization of the latent plasticity state during DBS.

## A.5 SHUFFLE CONTROLS: TESTING RELIABILITY AND IDENTIFIABILITY OF PLASTICITY INFERENCE.

We introduce two complementary shuffle experiments because there are two distinct failure modes when claiming *plasticity inference*. One is a **reliability/interpretability issue**: a high plasticity score could arise from static session-to-session heterogeneity even if there is *no coherent slow drift*. The other is an **identifiability issue**: the model class might *impose* smooth trends through its inductive bias, making it unclear whether the recovered slow trajectory is truly learned from the data.

### A.5.1 EVALUATION STAGE SHUFFLE (NULL DISTRIBUTION FOR THE PLASTICITY METRICS).

We keep the model trained on the true session order fixed, then generate many session-order-shuffled data and recompute exactly the same plasticity metrics. Because shuffling breaks coherent slow evolution while preserving within-session statistics, this provides a direct *chance baseline* for our plasticity readouts.

**Lorenz systems with parameter evolution.** Keeping the same trained model but evaluating it on data with a randomly shuffled session order (Appendix Fig. 3c) destroys the smooth and monotonic drift structure, yielding highly variable motif scales with no apparent trend, with DSA = 0.302,

which is larger than the smooth structure. Since baseline models lack predictive capability, we evaluate their performance using inferred embeddings rather than predicted ones. The resulting dynamical similarity analysis shows that baseline models exhibit a higher DSA value compared to STEER (Appendix Fig. 4b and c).

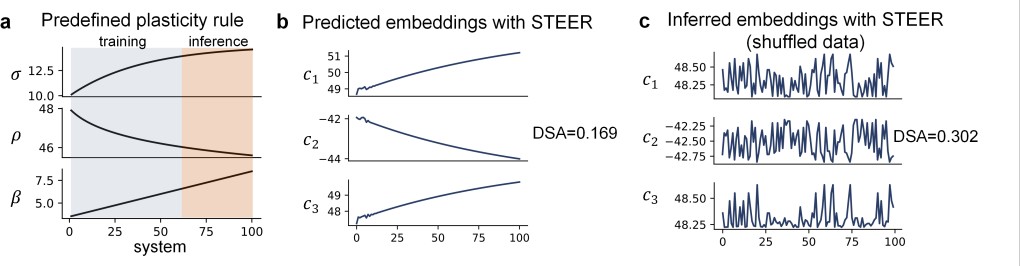

Appendix Fig. 3: Dynamical Similarity Analysis (shuffled data). (a) The predefined plasticity rule. (b) Predicted embeddings with STEER (no shuffle). (c) Inferred embeddings with STEER (random shuffled data).

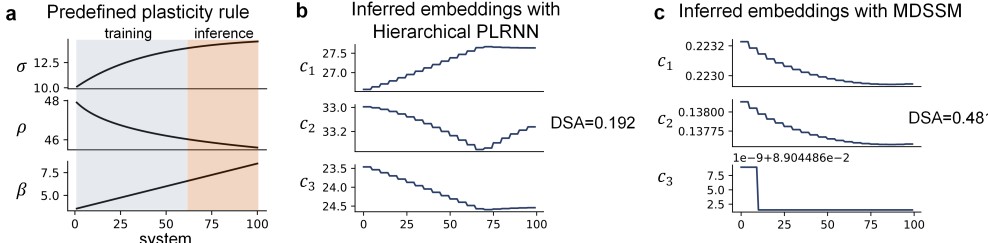

Appendix Fig. 4: Dynamical Similarity Analysis (baseline models). (a) The predefined plasticity rule. (b) Inferred embeddings with Hierarchical PLRNN. (c) Inferred embeddings with MD-SSM.

**Stimulus-evoked plasticity via BCM.** In this evaluation stage shuffle experiment, the trajectory of the motif scales $\mathbf{c}$ collapses under this evaluation-time shuffle (as in Appendix Fig. 5b). Also, the plasticity-related metrics of original order evaluation significantly exceed metrics in shuffled null distribution (Appendix Fig. 5c). The results indicate that the model's predictions rely on the real temporal organization rather than session-wise independent statistics.

**PD-DBS Longitudinal Neural Dataset.** To test whether the observed alignment between week-to-week changes in plasticity and FC could be explained by trivial factors, we performed a within-subject shuffle control. For each rat, we kept the trained model parameters fixed and generated surrogate datasets by randomly permuting the four weeks of data. For each shuffle, we re-computed the plasticity trajectory $\mathbf{c}$, projected it to 3D via PCA, and mapped it into the FC space using the Procrustes transform fitted on the original (unshuffled) data. Directional similarity was then quantified using the same cosine-based metric as in the main text. Appendix Fig. 6 summarizes this analysis across groups (PD, PD-DBS, sham). For all three groups, the real data (no shuffle) show substantially higher $\Delta\mathbf{c}$–$\Delta FC$ similarity than the week-shuffled baseline, indicating that the alignment cannot be explained by arbitrary rearrangements of weeks alone.

### A.5.2 TRAINING STAGE SHUFFLE (RULING OUT "THE MODEL HALLUCINATES DRIFT").

We also train the same architecture on session-order–shuffled data and ask whether it can still reproduce the ordered slow-trend patterns. If a similar trend emerged despite the absence of chronologi-

**a** Motif scale $c$ dynamics trained and evaluated by data in original session order

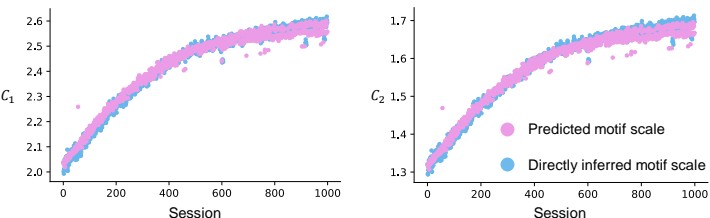

**b** Motif scale $c$ dynamics trained by data in original data and evaluated by shuffled session order

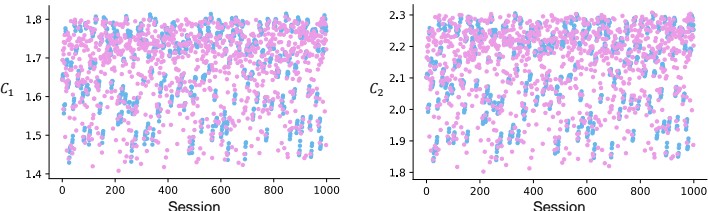

**c** Plasticity-related metrics of model evaluated by data in shuffled session order

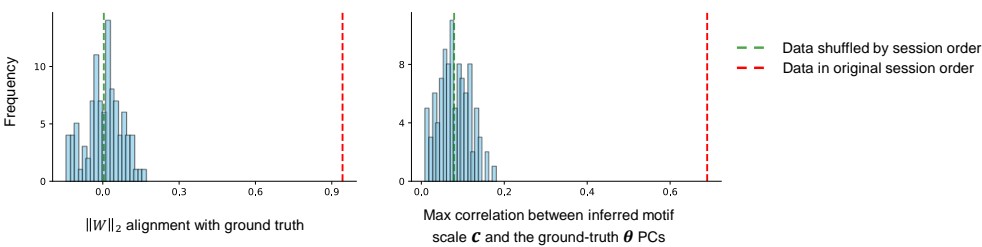

Appendix Fig. 5: Motif scales dynamics depend on the temporal ordering of sessions. (a) Predicted motif scales **c** (magenta) closely track the directly inferred motif scales **c** (blue) for two example motifs when the model is trained and evaluated on data in the original chronological session order, yielding smooth gradual changes across sessions. (b) Using the same model but evaluating on data with shuffled session order disrupts this structure, producing highly variable motif scales across sessions. (c) Original-order plasticity metrics significantly exceed the session-shuffled null distribution (t=-134, $p < 1e-5$ for $\|W\|_2$ alignment and t=-388, $p < 1e-5$ for max correlation).

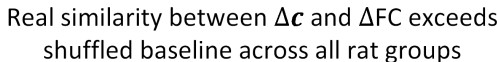

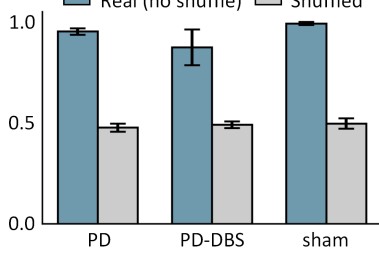

Appendix Fig. 6: Shuffle control for $\Delta$c–$\Delta FC$ alignment. Bars show the mean similarity between $\Delta$c and $\Delta FC$ across subjects ($\pm$ SEM) for each rat group (PD, PD-DBS, sham). Blue bars correspond to the real data (no shuffle); grey bars correspond to within-subject week-shuffled surrogate data. In all groups, the real similarity markedly exceeds the shuffled baseline, supporting that the model's plasticity embedding captures meaningful FC dynamics rather than artifacts of the embedding procedure.

cal structure, it would indicate that the trend is largely an artifact of model bias rather than inferred from temporal continuity. Instead, training on shuffled order removes the ordered slow-timescale trend, supporting that our slow plasticity dynamics are learned from coherent across-session structure present only in the correctly ordered data. Results in Appendix Tab. 1 and Appendix Fig. 7 support the claim that temporal ordering provides essential information (and an inductive bias) for capturing genuine cross-session slow dynamics rather than session-wise independent fitting or shortcut explanations.

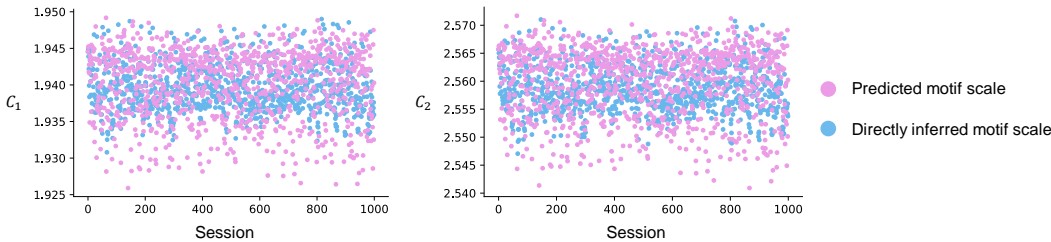

Appendix Fig. 7: Motif scales dynamics of model trained and evaluated by shuffled session order data in BCM synthetic dataset: Under this shuffled setting, the motif scales exhibit only stationary fluctuations across sessions and no longer show a systematic trend.

| Models | $\|W\|_2$ alignment ($\uparrow$) (across-session) | EV ($\uparrow$) (within-session) |
|---|---|---|
| STEER (original session order) | 0.9606±0.0036 | 0.9397±0.0008 |
| STEER (shuffled session order) | -0.0025±0.0132 | 0.9450±0.0007 |

Appendix Tab. 1: Performance of model training and evaluating by original and shuffled session order data in BCM synthetic dataset.

### A.6 JOINT TRAINING VS SEQUANTIAL TRAINING OF FAST AND SLOW DYNAMICS

In our model, the "fast" module learns a set of cross-session motifs, and the "slow" module captures how these motifs are expressed over longer timescales. Training fast dynamics first and then fitting the slow dynamics on top of fixed motifs is therefore equivalent to optimizing the same objective in two sequential parameter blocks (fast, then slow), rather than jointly. If the motifs are not held fixed in the second stage, this procedure effectively reduces to standard joint optimization and the distinction between "fast" and "already-learned" components becomes moot. A direct comparison of the two training procedures in the BCM experiment is shown in Appendix Tab. 2 and Appendix Fig. 8. Across sessions, both approaches produce very similar trajectories of the inferred motif scales, indicating that the learned slow dynamics are largely unaffected by whether fast and slow components are optimized jointly or in sequence. Joint training yields slightly smaller discrepancies at early and late sessions, but the overall across-session trends and 5-session-ahead forecasts are essentially indistinguishable between the two schemes.

| Models | $\|W\|_2$ alignment (across-session) | EV (within-session) | Correlation between $\mathbf{c}$ and $\mathbf{c}_{pred}$ |
|---|---|---|---|
| STEER with jointly learning | 0.9606±0.0036 | 0.9397±0.0008 | 0.9934 ± 0.0016 |
| STEER with staged learning | 0.9602±0.0076 | 0.9310±0.0022 | 0.9866 ± 0.0094 |

Appendix Tab. 2: Model performance by different training procedures in BCM synthetic dataset.

### A.7 HYPERPARAMETER EVALUATION

**Lorenz systems with parameter evolution.** We examined the sensitivity of the model to learning rate, tensor rank, hidden dimension, and the regularization parameter $\lambda_{slow}$ ($\lambda_{smooth}$ is the same value as $\lambda_{slow}$). Here, we evaluate the predictive performance of our model. For the fast time

**a**    Motif scale *c* dynamics by jointly training fast and slow dynamics

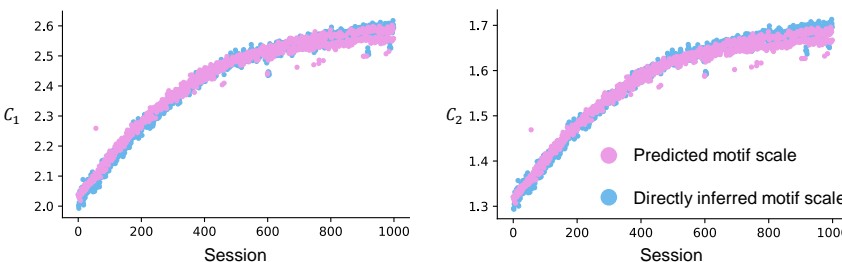

**b**    Motif scale *c* dynamics by sequential training fast and slow dynamics

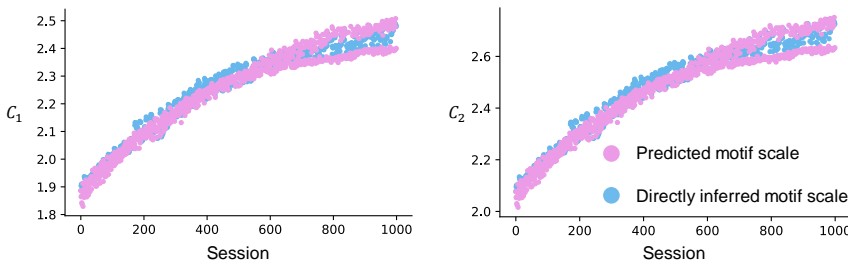

Appendix Fig. 8: Motif scales dynamics by two training schemes in BCM synthetic dataset. (a) Session-by-session evolution of motif scales when fast and slow dynamics are trained jointly. Curves show 5-session-ahead forecasts of the slow dynamics (predicted motif scales) compared to directly inferred motif scales. (b) Same as in (a), but with sequential training where fast dynamics are learned first with fixed motifs across sessions.

scale, we used EV to evaluate the predictive performance. The results in Appendix Fig. 9 indicate that, within a wide range, the choice of these hyperparameters has little impact on short-timescale prediction.

**Stimulus-evoked plasticity via BCM.** We examined the sensitivity of the model to learning rate, tensor rank and the regularization parameter $\lambda_{slow}$ (we do not consider $\lambda_{smooth}$ in this experiment). Here, we evaluate our model on two timescales. For the fidelity of the inferred slow law, we used max correlation between PCs of $\theta$ and inferred motif scales $\mathbf{c}$, and the alignment of the inferred $\|\mathbf{W}\|_2$ dynamics with the ground truth. For the fast time scale, we used EV to evaluate the predictive performance. The results in Appendix Fig. 10 indicate that, within a wide range, the choice of these hyperparameters has little impact on either short-timescale prediction or on the accuracy of inferred plasticity. In contrast, varying the number of forecasting sessions revealed a dissociation between short-term prediction and long-term plasticity inference. With longer forecasting horizons, within-session stayed high while both the max correlation and $\|\mathbf{W}\|_2$ dynamics alignment declined systematically. Thus, extending the forecasting window preserves short-term behavioral performance of the model but compromises its ability to faithfully reconstruct the latent plasticity dynamics.

**Task Learning through Stimulation-Induced Plasticity.** To assess robustness on the task learning dataset, we varied the rank of the shared low-rank latent state space, the learning rate and $\lambda$. In Appendix Fig. 11, the rank-7 model is the best one to approximate the task-learning dynamics. The results in Appendix Fig. 11b and c indicate that, within a wide range of learning rates, $\lambda$ has little impact on weight similarity.

**PD-DBS Longitudinal Neural Dataset.** To assess robustness on the PD-DBS dataset, we varied the rank, the hidden dimension of the shared low-rank latent state space, and the regularization parameter $\lambda_{slow}$ ($\lambda_{smooth}$ is the same value as $\lambda_{slow}$), over the ranges shown in Fig. 12. For each configuration, we evaluated within-session predictive performance (EV) and the similarity between inferred coupling changes $\Delta\mathbf{c}$ and observed functional connectivity changes $\Delta FC$. Across this

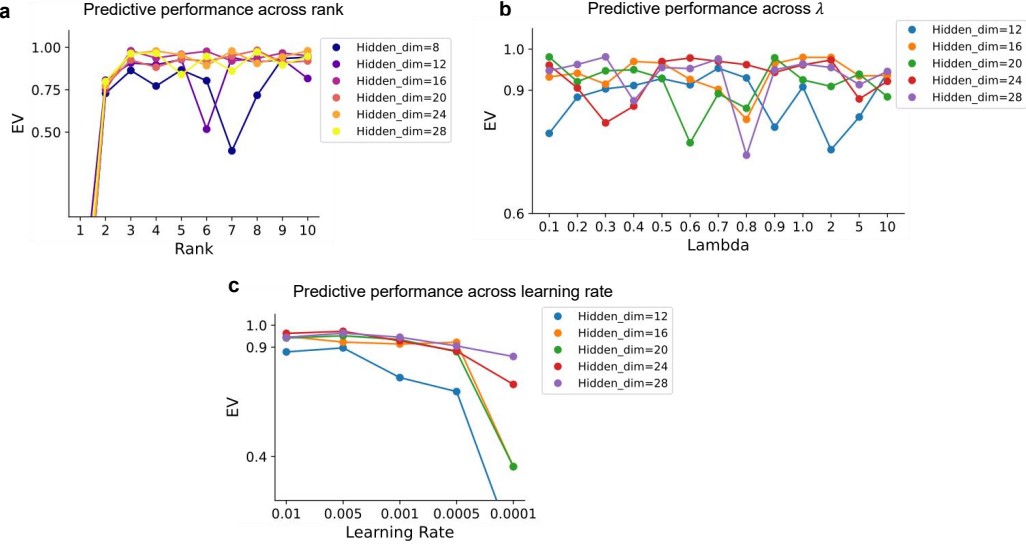

Appendix Fig. 9: Predictive performance across different hyperparameters. (a) Predictive performance across ranks under different hidden dimensions (rank=3 is stable to explain most of the variance in most of the hidden dimension settings). (b) Predictive performance across $\lambda$ under different hidden dimensions at rank=3. (c) Predictive performance across learning rates at rank=3 (A higher hidden dimension has stable predictive performance across learning rates).

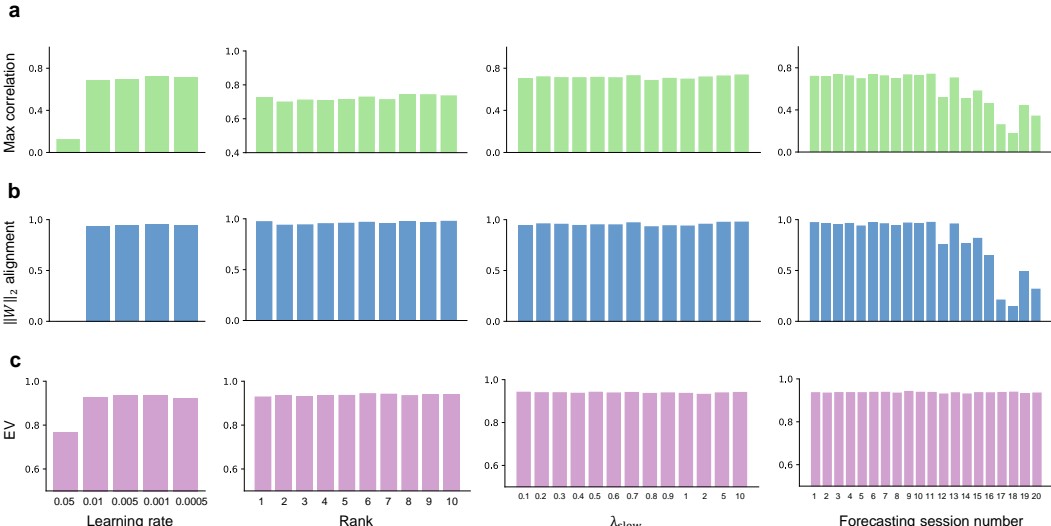

Appendix Fig. 10: Robust model performance across hyperparameters for inferring stimulus-evoked plasticity via BCM. (a) Max correlation across hyperparameters between ground-truth sliding thresholds $\theta$ PCs and inferred motif scales **c**. (b) Normalized L2-norm of **W** dynamics alignment across hyperparameters between the ground truth and the inferred dynamics. (c) Within-session predictive performance (EV) across hyperparameters.

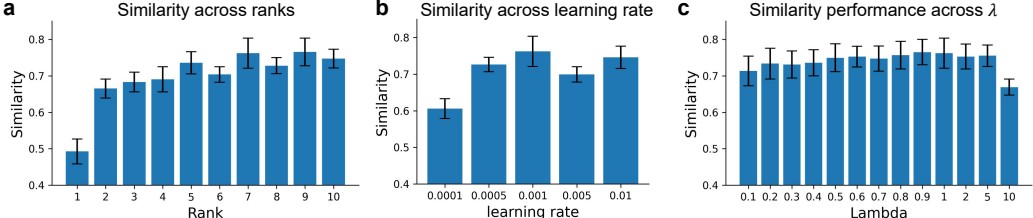

Appendix Fig. 11: Similarity of cycle-wise W between model inference and ground truth across ranks (a), learning rate (b) and $\lambda$ (c).

sweep, both metrics remain consistently high and change only modestly, indicating a broad plateau of good performance. The hyperparameters used in our main PD-DBS experiments lie well within this stable region. Thus, our conclusions on the PD-DBS data do not rely on fine-tuning: STEER reliably reconstructs neural activity and recovers the relationship between slow coupling changes and functional connectivity across a wide range of settings.

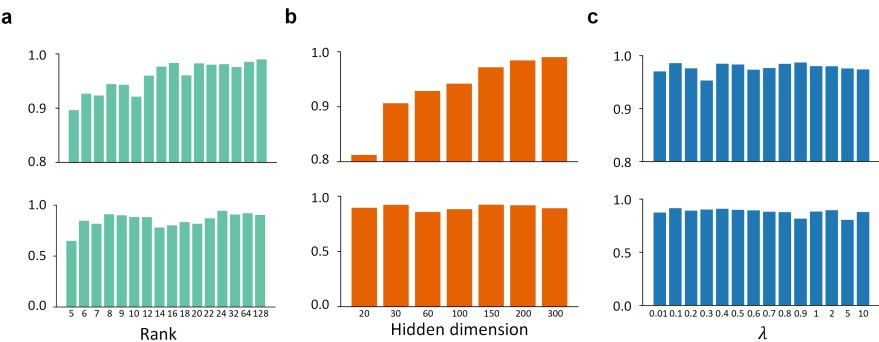

Appendix Fig. 12: Robust model performance across hyperparameters on the PD-DBS dataset. Each column varies one hyperparameter while keeping the others fixed: **(a)** Tensor rank of the connectivity factorisation, **(b)** The hidden dimension in the shared low-rank latent state space, and **(c)** Smoothness/consistency regularisation coefficient $\lambda$. Top row: Within-session predictive performance (EV) across hyperparameters. Bottom row: similarity between inferred changes in recurrent coupling $\Delta\mathbf{c}$ and observed changes in functional connectivity $\Delta FC$.

## A.8 IMPLEMENTATION DETAILS

**Lorenz dynamics.**

- Low-rank RNN: RNN (28), rank 3.
- Plasticity Predictor: RNN (28)
- State Encoder: MLP (28, 28)
- State Decoder: Linear Projection (28 → 3)
- Plasticity Encoder: MLP (28, 28)
- Weight Inference: Linear Projection (28 → 3)

**Stimulus-evoked plasticity via BCM.**

- Low-rank RNN: RNN (50), rank 2.
- Plasticity Predictor: RNN (10)
- State Encoder: MLP (50, 50)

- State Decoder: Linear Projection ($50 \rightarrow 50$)
- Plasticity Encoder: GRU (50)
- Weight Inference: Linear Projection ($10 \rightarrow 2$)

**Task learning through stimulation-induced plasticity.**

- Low-rank RNN: RNN (50), rank 7.
- Plasticity Predictor: RNN (50)
- State Encoder: MLP (50, 50)
- State Decoder: Identity Matrix
- Plasticity Encoder: MLP (50,50)
- Weight Inference: Linear Projection ($50 \rightarrow 7$)

**PD-DBS.**

- Low-rank RNN: RNN (200), rank 24.
- Plasticity Predictor: RNN (200)
- State Encoder: MLP (200, 200)
- State Decoder: Linear Projection ($200 \rightarrow$ channel)
- Plasticity Encoder: MLP (200, 200)
- Weight Inference: Linear Projection ($200 \rightarrow 24$)

All networks are trained with the Adam optimizer, and the learning rate is tuned by selecting the best predictive performance. We initialize parameters using Xavier normal, Xavier uniform, and zero-mean Gaussian initializations; remaining nn.Linear weights follow PyTorch's default initialization (with optional orthogonality constraints via GeoTorch).

### A.9 USE OF LARGE LANGUAGE MODELS (LLMS)

To enhance the overall quality and presentation of the manuscript, we made limited use of a large language model (LLM) during the writing process. LLM assisted us in refining wording, improving sentence clarity, and checking grammar and narrative fluency. In addition, it provided support in organizing certain sections' logical structure and narrative flow, helping us present our research in a clearer, more coherent, and more persuasive manner.

It is important to note that the LLM's involvement was strictly confined to textual polishing and structural expression. The study's core ideas, research design, experimental procedures, data analysis, and scientific conclusions were conceived and carried out entirely by the authors. All content of the manuscript, including any suggestions generated by the model, was thoroughly reviewed, edited, and verified by the authors, who take full responsibility for the scientific rigor, accuracy, and originality of the final work.

