# OpenReview forum: "Inferring brain plasticity rule under long-term stimulation with structured recurrent dynamics"
_ICLR.cc/2026/Conference — ICLR 2026 Poster_

### Official Review · Reviewer_YUkh · 2025-10-29

**Soundness:** 3
**Presentation:** 2
**Contribution:** 4
**Rating:** 4
**Confidence:** 3

**Summary:**

The original idea and intuition behind this paper are excellent.

STEER represents recurrent connectivity as a low-rank CP tensor decomposition and learns a stimulus-conditioned dynamical law $z_{k+1}=g_{\theta}(z_k,\bar{u}_k)$  governing how low-dimensional motif coefficients evolve across sessions. Identifiability is promoted through unit-norm and orthogonality constraints and an optional sign mask enforcing Dale’s principle. The approach is validated on increasingly realistic tasks: synthetic Lorenz systems, a Bienenstock–Cooper–Munro (BCM) plasticity model, and a longitudinal Parkinson’s disease DBS dataset.

Several essential details are relegated to the appendix and the presentation undermine the overall quality of the work. Some of those choices make me dubious on the validity of the model.

I am giving the paper an overall 4 "marginally below the acceptance threshold" but will consider increasing that grade if the weaknesses are adresses.

**Strengths:**

This work is well motivated and conceptually strong and addresses a fundamental challenge in neuroscience of understanding how long-term stimulation reshapes neural circuits. The proposed framework introduces a dual-timescale formulation that separates fast within-session dynamics from slow plasticity adaptation.

The paper is innovative. By employing a low-rank decomposition of the recurrent connectivity tensor, STEER enforces structure and identifiability, leading to interpretable motif-level representations of plasticity.

The experimental design is particularly strong. The authors carefully structure their validation in increasing order of biological realism: starting from the synthetic Lorenz system, progressing to a controlled Bienenstock–Cooper–Munro (BCM) plasticity model, and culminating with real longitudinal deep-brain stimulation (DBS) data in Parkinsonian rats.

**Weaknesses:**

The presentation needs a brush

Clarity:
-"session" is never really well defined in the beginning and is such a vague term. Does itrefers to a single day, trial block, animal or experiment ?
- section 4.1. Lorentz equation should be in main text, not hidden in the appendix
- l299-204.  the are no motivation behind equation 10
- Figure 4 is too small

Motivation
-  In the Lorentz experiment, it's not clear why this specific synthetic plasticity is chosen; they appear arbitrary.  A brief justification would be okay and would help readers evaluate whether the benchmark reflects plausible plasticity.  (you already justify parameters range, does the same justification work for the function form ?)

Conceptually:
- you are presenting the performances on the states but the evaluation of plasticity (evaluation of the parameters) changes from benchmark to benchmark
- similarly you don't provide systematic comparison with other methods

**Questions:**

* Identifiability and factor uniqueness: The authors reduce CP scaling/sign indeterminacy by constraining motif factors to unit‑norm and penalising orthogonality.  Do these constraints guarantee a unique solution, or could different factor orderings or scalings still yield the same connectivity?  How do they interpret the motifs biologically when multiple equivalent factorizations exist?

* Report performance on the inferred plasticity, not just on state prediction. The paper emphasises within‑session predictive accuracy ($R^2$) and explained variance on held‑out trajectories, but these metrics speak only to how well the model predicts neural activity.  For a method whose main goal is to infer the slow plasticity rule, one would also expect quantitative assessments of how accurately the latent plasticity dynamics $z_{k+1}=g_\theta(z_k,\bar{u}_k)$ and the motif coefficients $c_k$ recover the ground truth.  In the Lorenz and BCM benchmarks, the authors report a dynamical‑similarity score (DSA = 0.63) and correlations with BCM thresholds, but there is no systematic evaluation of plasticity inference across rank choices or hyperparameters.  Likewise, in the DBS experiment they assess alignment between $\Delta c$ and functional connectivity rather than the fidelity of the inferred slow law.  Expanding these analyses to provide error metrics on the inferred plasticity would strengthen the claim that STEER recovers the underlying rule.

---

> ### Author Response · Authors · 2025-11-24
> **Response to Reviewer YUkh (1/3)**
>
> We thank the reviewer YUkh considering that **the original idea and intuition behind this paper are excellent**.
>
> We have addressed the concerns and questions point-by-point below, and have incorporated the corresponding changes in the revised version.
>
> # W1: Presentation and clarity
>
> Thanks for your advice, in our revised paper, we have tried to make our
> presentation more readable and clearer. Meanwhile, the Lorenz equation
> has added to the main text.
>
> # W2: Motivation of the Lorenz plasticity choice
>
> In our design of the synthetic plasticity rule, we aimed to model a
> plausible dynamic process that mirrors what is observed in biological
> neural networks, where synaptic connections can be strengthened or
> weakened depending on various factors. Specifically, we chose to model
> two parameters increasing while one decays, which we believe reflects
> the real-world balance between potentiation and depression in neural
> connections. In the brain, such processes are often complementary, where
> certain synaptic connections are reinforced, while others are suppressed
> to optimize learning and network behavior.
>
> Furthermore, we chose the Lorenz system as an initial benchmark due to
> its well-established reputation as a representative model for dynamical
> systems. The simplicity of this benchmark allows us to assess our
> plasticity rule in a controlled setting. We believe this starting point
> is important for demonstrating the core dynamics of our model, and it
> serves as a foundation for later extending the rule to more biologically
> relevant benchmarks involving complex plasticity mechanisms. This
> step-by-step approach allows us to gradually scale up the model to more
> intricate biological systems while ensuring the validity of the
> underlying principles.
>
> # W3: Conceptual concerns: evaluation metrics and baselines
>
> Long-term plasticity, however, is an implicit and relatively abstract
> concept in our setting: it is not given as a single explicit closed-form
> rule that is shared across all datasets. Instead, each experimental
> setup exposes a *different observable manifestation* of the underlying
> slow plasticity rule, and we therefore tailor the evaluation metric to
> the aspect that is actually identifiable in that system.
>
> Concretely:
>
> -   **Lorenz benchmark.** Here we explicitly design slowly varying
>     parameters governing the recurrent dynamics. In this case, the
>     "plasticity rule" is precisely the evolution of these parameters
>     over sessions, so we evaluate whether STEER recovers their
>     session-wise trajectories and similarity between the ground truth
>     plasticity rule and scaling factors.
>
> -   **BCM benchmark.** In this model, repeated stimulation induces
>     changes in the recurrent weights $W$. The trend of $W$ as a function
>     of stimulation history is the key expression of the plasticity rule.
>     Accordingly, we focus on whether STEER captures the correct
>     dependence of $W$ on the stimulation protocol. Meanwhile, we also
>     evaluate the correlation between the scaling factor and the
>     evolution of $\theta$ in BCM rules, since the dynamics of $\theta$
>     influence the network plasticity.
>
> -   **PD-rat dataset.** For the real DBS recordings, long-term
>     plasticity is reflected in slow changes of functional connectivity
>     across sessions rather than in directly observable synaptic
>     parameters. In this case, the most meaningful observable proxy for
>     the plasticity rule is the trajectory of functional connectivity
>     motifs, and our evaluation therefore concentrates on how well STEER
>     recovers these trends.
>
> Although the numerical metrics differ, they are conceptually aligned: in
> all three benchmarks, we assess whether STEER recovers the *structure
> and direction of long-term changes* induced by stimulation, which is
> where the slow plasticity rule manifests itself in each system. Using a
> single unified metric across all datasets would require abstracting away
> these system-specific observables and might obscure the biologically
> relevant aspects of plasticity.
>
> Regarding systematic comparison with other methods, in each simulated
> benchmark, we evaluate STEER side-by-side with the baseline models
> introduced in the main text (e.g., MD-SSM and hierarchical PLRNN), using
> the same training protocol, data splits, and task-specific metrics. This
> ensures that any performance differences reflect how well each model
> captures the relevant long-term plasticity, rather than artifacts of the
> evaluation procedure. Thus, although the observables and metrics are
> tailored to each experimental setting, the comparisons with alternative
> methods are systematic within each benchmark.

---

> ### Author Response · Authors · 2025-11-24
> **Response to Reviewer YUkh (2/3)**
>
> # Q1: Identifiability and factor uniqueness
>
> **Do the constraints guarantee a unique factorization?** Strictly
> speaking, no: the constraints we impose (unit-norm columns in $A,B$,
> orthogonality penalties, and optional Dale's mask) reduce the usual CP
> indeterminacies but do not mathematically guarantee a unique $(A,B,C)$
> up to the standard symmetries. Constraining $\\|a\_r\\|\_2 = \\|b\_r\\|\_2 = 1$
> and absorbing scales into $c\_{r}^k$ removes arbitrary rescaling, and the
> orthogonality penalty $\\|A^\top A - I\\|\^2 + \\|B^\top B - I\\|^2$
> discourages highly redundant motifs. As in most CP models, there
> remains:
>
> -   permutation symmetry across components $r$;
>
> -   sign symmetry at the level of $a\_r, b\_r$ (flipping both leaves
>     $a\_rb\_r^\top$ unchanged);
>
> -   potential non-uniqueness if two motifs span nearly the same
>     subspace.
>
> In addition to reducing these indeterminacies, the constraints have a
> *practical optimization benefit*: they make the optimization landscape
> better conditioned by avoiding arbitrarily large/small scales across
> factors and by discouraging component collapse. Empirically, this leads
> to faster and more stable convergence across random initializations.
>
> What we *empirically* find is that the optimization converges to
> qualitatively similar and quantitatively well-aligned sequences of
> effective connectivity operators $\{W\_k\}$ in the *latent* dynamical
> space (up to the usual low-rank symmetries and small variability across
> local minima).
>
> **Can different orderings/scalings yield the same connectivity?** Yes,
> different permutations of the motifs, together with the corresponding
> permutation of the columns of $C$ and the parameters of $g\_\theta$,
> describe the same family of latent operators $W\_k$ and the same
> trajectories in connectivity space, expressed in a relabeled coordinate
> system. Similarly, global rescalings of $(a\_r,b\_r,c\_{r}^k)$ that respect
> the unit-norm constraints are effectively absorbed into $c\_{r}^k$. In
> that sense, STEER identifies the *geometry* of plasticity in a low-rank
> latent connectivity space (the trajectory of $W\_k$ and of the motif
> coefficients), but not a unique naming of components.
>
> **How we interpret motifs biologically given equivalent
> factorizations?** Crucially, in our framework the motifs live in a
> *latent* space: they parameterize low-rank structure of an effective
> connectivity operator acting on latent population states. Biologically,
> we therefore interpret motifs at the level of *system-level neural
> dynamics*. That is, in terms of how coordinated patterns of interaction
> between neural populations reorganize across disease states and under
> stimulation, rather than as single identified synapses or tracts. Our
> interpretation focuses on quantities that are invariant or robust under
> the remaining symmetries:
>
> 1.  **Latent, dynamical structure with system-level biological
>     meaning.** In our setting, a single motif $a\_r b\_r^\top$ is best
>     viewed as a direction along which the *latent dynamics* can
>     reorganize under different stimuli or conditions, rather than as a
>     specific anatomically labeled circuit. Each motif captures a
>     coordinated pattern of change in the effective connectivity operator
>     $W\_k$ that shapes biologically relevant dynamical properties in the
>     latent space. For example, how strongly and how quickly population
>     activity responds to perturbations, how persistent pathological
>     activity patterns are, or how fast plastic changes accumulate and
>     saturate over days or weeks. Accordingly, in the present work we
>     interpret motifs through their dynamical role, for example, by
>     examining how trajectories of the motif coefficients and of $W\_k$
>     evolve and slow down across sessions or conditions, and how these
>     trajectories relate to changes in functional connectivity and
>     behavioral recovery, rather than through a one-to-one mapping onto
>     physical microcircuits.
>
> 2.  **Scale-dependent interpretation.** The same STEER formalism can in
>     principle be applied at different observational scales: from local
>     population activity to whole-brain or multi-area latent embeddings.
>     In all cases, motifs are interpreted as directions of coordinated
>     change in the latent dynamics at that scale (e.g. reweighting of
>     interactions between recorded neurons, or between large-scale
>     networks). Biologically, this corresponds to describing how
>     *network-level interactions* reorganize under disease and
>     stimulation.

---

> ### Author Response · Authors · 2025-11-24
> **Response to Reviewer YUkh (3/3)**
>
> # Q2: Reporting performance on inferred plasticity (beyond state prediction)
>
> We agree that a more direct evaluation of how well STEER recovers the
> inferred plasticity dynamics is essential, especially given the primary
> focus of our method on uncovering the underlying slow plasticity rule.
>
> To address this concern, we have conducted additional analyses to assess
> the accuracy of the latent plasticity dynamics and motif coefficients
> across different benchmark tasks under different hyperparameter settings
> (see Appendix Fig.3-10). Specifically:
>
> -   **Lorenz and BCM benchmarks**: We include a more systematic
>     evaluation of the plasticity inference across rank choices and
>     hyperparameter settings. This involves computing error metrics
>     (e.g., explained variance, dynamical similarity scores, or
>     correlation coefficient) to quantify how closely the inferred
>     plasticity dynamics and motif coefficients match the ground truth
>     values. This analysis reflects the robustness of our framework.
>
> -   **DBS experiment**: We acknowledge that our current evaluation
>     focuses more on the alignment between neural activity and functional
>     connectivity. To further evaluate the robustness of our model, we
>     expand this analysis by introducing a direct assessment of the
>     fidelity of the inferred slow plasticity law under shuffled data.
>     Specifically, we shuffle the data to assess whether the model is
>     learning meaningful plasticity dynamics or merely exploiting noise
>     in the data. We then compare the predicted plasticity dynamics with
>     the known ground truth and report quantitative metrics to evaluate
>     the accuracy of the inference under shuffled conditions (see
>     Appendix Fig.6). This analysis helps ensure that the model is not
>     overfitting to idiosyncratic features of the dataset and is instead
>     capturing biologically relevant dynamics.
>
> These expanded analyses provide a more comprehensive assessment of
> STEER's ability to recover the underlying plasticity rule and strengthen
> our claim that the model can accurately infer slow, long-term changes in
> neural dynamics.

---

> > ### Comment · Reviewer_YUkh · 2025-11-26
> >
> > Dear author,
> >
> > Thank you for the rebuttal. I will take the time to read it carefully and aim to share my feedback before the end of the discussion period.
> >
> > Best regards,

---

> ### Author Response · Authors · 2025-11-27
> **Additional response to Reviewer YUkh**
>
> Dear Reviewer,
>
> Thank you for your message and for taking the time to read our rebuttal
> carefully. To make our responses easier to follow, we have added a
> concise summary table that highlights the plasticity-related and state
> prediction performance of our model, comparing to the baselines. We hope
> this table helps you quickly locate the changes.
>
> | Models | Predictive performance (EV ↑) | Dynamical similarity (DSA ↓) |
> |---|---:|---:|
> | Low-rank MD-SSM (rank = 3) | 0.4891±0.1004 | 0.1700±0.0001 |
> | Full-rank MD-SSM | 0.7774±0.0646 | 0.1598±0.0176 |
> | Hierarchical-PLRNN (full rank) | 0.9313±0.0569 | 0.1519±0.0245 |
> | **STEER without forecasting (rank = 3)** | **0.9575±0.0236** | 0.1694±0.0001 |
> | **STEER forecasting all sessions (rank = 3)** | 0.9471±0.0165 | **0.1301±0.0483** |
> *Table: Model performance across random seeds in Lorenz synthetic dataset.*
>
> On the Lorenz synthetic dataset, STEER consistently outperforms all
> baselines on both short-term predictive accuracy (EV) and long-term
> plasticity-related dynamical similarity (DSA), indicating superior
> performance in both prediction and preserving long-horizon dynamics.
>
> | Models | $\lVert W \rVert_2$ alignment (across-session) | EV (within-session) |
> |---|---:|---:|
> | Low-rank MD-SSM without forecasting (rank 2) | 0.1897±0.2250 | 0.9348±0.0111 |
> | Full-rank MD-SSM without forecasting| 0.0053±0.1932 | 0.9502±0.0048 |
> | Hierarchical-PLRNN without forecasting (full-rank) | 0.9743±0.0041 | 0.9479±0.0013 |
> | **STEER without forecasting (rank 2)** | 0.9706±0.0038 | 0.9460±0.0012 |
> | **STEER with forecasting 5 sessions (rank 2)** | 0.9606±0.0036 | 0.9397±0.0008 |
>
> *Table: Model performance across random seeds in BCM synthetic dataset.*
>
> On the second dataset, STEER retains a low-rank structure while enabling
> long-term forecasting, yet still matches other models in short-term EV
> and achieves plasticity-related performance comparable to the full-rank,
> no-forecasting hierarchical PLRNN model adapted to fit our settings.
>
> Taken together, these results suggest that STEER offers a favorable
> trade-off between efficiency and the ability to capture long-term
> plasticity-related trends, without sacrificing short-term predictive
> accuracy.
>
> Thank you again for your consideration.
>
> Best regards,

---

### Official Review · Reviewer_sTJk · 2025-10-29

**Soundness:** 3
**Presentation:** 3
**Contribution:** 3
**Rating:** 6
**Confidence:** 3

**Summary:**

This paper developed a framework, named Stimulus-Evoked Evolution Recurrent dynamics (STEER), a structured model that separates fast within-session activity from slow across-session adaptation. The method explored how recurrent connectivity evolves under repeated interventions.

**Strengths:**

1. The paper treats long-horizon plasticity as a latent dynamical law, rather than unstructured parameter drift.
2. The model separates fast within-session dynamics and slow across-session evolution.

**Weaknesses:**

1. The dynamical systems have input weights and readouts, which can also encode some information of connectivity. Therefore, it is still unsure if the connectivity recovered by the model is true or believable. (This may have been claimed by the author in the limitation part, but it remains a substantive concern.)
2. The authors stated that the proposed method enforces an identifiable separation between fast within-session responses and slow network reconfiguration. But there’s no theorem or ablation demonstrating uniqueness of learned dynamics.

**Questions:**

1. If you learn fast dynamics at first, then learn slow dynamics of these fast ones, will the results be different? Learning jointly requires hyperparameter search such as lambda slow, are the results sensitive to these parameters?
2. Could the author define delta W? (Figure 4D) Since there are many Ws in the paper.

---

> ### Author Response · Authors · 2025-11-24
> **Response to Reviewer sTJk (1/3)**
>
> We thank the reviewer for the thoughtful comments and for recognizing
> the conceptual contribution of separating fast within-session activity
> from slow across-session adaptation. Below we address the main concerns
> and questions in detail.
>
> # W1: Interpretability and "truth" of recovered connectivity
>
> We agree that, in general, connectivity in recurrent dynamical systems
> is only identifiable up to certain re-parameterizations and that input
> and readout weights can in principle absorb parts of the effective
> interactions. Our goal is therefore not to recover the true
> connectivity, but to identify a *low-dimensional, protocol-dependent
> plasticity law and motif-level connectivity patterns* that are as
> constrained as possible by the data and the model architecture.
>
> In our model we take several steps to reduce the degeneracy the reviewer
> is concerned about:
>
> 1.  **Only the recurrent connectivity changes across sessions.** The
>     input matrix $W_{\text{in}}$ and the linear readout are shared
>     across sessions and do not depend on the plasticity embedding $z_k$
>     or on the stimulation protocol. In contrast, the recurrent weight
>     $W_k$ is the only session-varying parameter, and it is entirely
>     determined by the low-dimensional coefficients $c_k$ and the
>     plasticity law $z_{k+1} = g_\theta(z_k, \bar u_k)$. Consequently,
>     any systematic, protocol-dependent change across sessions cannot be
>     absorbed by the input or readout weights; it must be captured in
>     $W_k$ (via $c_k$ and $z_k$).
>
> 2.  **Structured low-rank motifs and identifiability constraints.** We
>     parametrize $W_k$ using a low-rank decomposition with shared spatial
>     motifs $a_r, b_r$ across all sessions and session-specific scaling
>     coefficients $c_{k,r}$. We constrain $\\|a_r\\|_2 = \\|b_r\\|_2 = 1$,
>     penalize deviations from orthogonality of $A$ and $B$, and
>     optionally impose an excitatory/inhibitory sign mask(In the BCM
>     experiment). These constraints reduce scaling and rotation
>     indeterminacies and force the model to explain slow changes through
>     a small set of shared motifs whose strengths evolve in a controlled
>     way, rather than through arbitrary combinations of input and output
>     weights.
>
> 3.  **Empirical validation of "believability" on ground-truth systems.**
>     In our synthetic experiments, both the true slow rule and the
>     ground-truth evolution of the recurrent weights are known. Under the
>     same architecture (including input and readout), the model:
>
>     -   recovers motif scaling trajectories $c_k$ whose evolution
>         matches the predefined parameter law in a dynamical-systems
>         benchmark, and
>
>     -   produces session-by-session changes in $\\|W_k\\|$ that closely
>         track the ground truth generated by a known plasticity rule,
>         with motif coefficients $c_k$ that strongly correlate with the
>         principal components of the true latent control variables.
>
>     These results show that, despite the presence of input and readout
>     weights, the model assigns the slow, protocol-dependent
>     reconfiguration to $W_k$ in a way that is quantitatively consistent
>     with the underlying plasticity mechanism when the ground truth is
>     available.
>
> 4.  **Real data: interpreting motifs as effective connectivity, not
>     literal synapses.** For experimental datasets, the true synaptic
>     matrix is of course unknown. Accordingly, we are cautious in our
>     claims: we interpret $W_k$ and $c_k$ as *effective recurrent motifs*
>     rather than exact anatomical connectivity. The fact that the
>     inferred motif trajectories align with changes in empirical
>     functional connectivity across sessions suggests that they capture
>     meaningful circuit-level reorganizations, rather than arbitrary
>     shifts in the readout.

---

> ### Author Response · Authors · 2025-11-24
> **Response to Reviewer sTJk (2/3)**
>
> # W2: "Identifiable separation" and uniqueness of the learned fast/slow dynamics
>
> We agree that our current manuscript does not provide a formal theorem
> guaranteeing uniqueness of the learned dynamics in the sense of global
> structural identifiability.
>
> What we mean by an "identifiable separation between fast within-session
> responses and slow network reconfiguration" is a structurally enforced
> and empirically testable factorization, rather than a proof that no
> alternative parametrization can ever explain the data equally well.
>
> Concretely:
>
> -   **Architectural separation of timescales.**
>
>     -   All within-session neural dynamics are generated by a single
>         shared low-rank RNN; session-to-session variability enters only
>         through the recurrent connectivity $W_k$.
>
>     -   The connectivity tensor $\{W_k\}$ is constrained to a CP
>         low-rank form with shared motif factors $A, B$ and
>         session-dependent motif scales $c_k$, together with norm
>         constraints, orthogonality penalties, and optional E/I sign
>         masks. Thus, the only degree of freedom for slow change is the
>         low-dimensional coefficient trajectory $c_k$.
>
> -   **Slow plasticity is confined to a low-dimensional law.**
>
>     -   Session-level plasticity embeddings $z_k$ are inferred from data
>         and evolve according to a learned recurrence
>         $z_{k+1} = g_\theta(z_k, \bar u_k)$; the motif scales are a
>         linear readout $c_k = M z_k + b_c$.
>
>     -   The training loss includes (i) a within-session multi-step
>         prediction term $L_{\text{fast}}$, (ii) a law-consistency term
>         $L_{\text{slow}} = \sum_k \lVert z_{k+1} - g_\theta(z_k, \bar u_k) \rVert^2$,
>         and (iii) a total-variation penalty on $c_k$ across sessions.
>         These terms explicitly discourage explanations in which slow
>         changes are absorbed into high-dimensional fast parameters, and
>         instead force the slow rule $g_\theta$ to account for
>         cross-session evolution.
>
> -   **Empirical evidence for practical identifiability.**
>
>     -   On the Lorenz benchmark, we define a known session-dependent
>         parameter evolution and show that STEER recovers a
>         low-dimensional scaling trajectory whose temporal structure
>         closely matches the ground-truth rule (low DSA), while baselines
>         (MD-SSM, hierarchical-PLRNN) fit the data but exhibit worse
>         dynamical similarity.
>
>     -   On the BCM plasticity benchmark, when a ground-truth synaptic
>         plasticity rule is used to generate data, STEER not only
>         achieves comparable within-session predictive performance to
>         MD-SSM, but also much better recovers the across-session
>         evolution of $\lVert W_k \rVert_2$ and yields motif scales that
>         correlate strongly with the principal components of the true
>         sliding thresholds $\theta_k$.
>
>     -   In an additional shuffle control where the session order is
>         randomly permuted, the smooth, monotonic drift of the inferred
>         scales disappears and the DSA score worsens, whereas the
>         original ordered data preserve a structured trajectory. This
>         ablation shows that the slow latent evolution learned by STEER
>         is tied to the true temporal ordering and not to static
>         heterogeneity.
>
> Together, these design choices and experiments support a practically
> identifiable factorization of fast within-session dynamics and slow,
> protocol-dependent plasticity: the slow rule lives in a restricted,
> low-dimensional hypothesis class and is constrained by both
> cross-session consistency and out-of-distribution protocol prediction.

---

> ### Author Response · Authors · 2025-11-24
> **Response to Reviewer sTJk (3/3)**
>
> # Q1: Joint vs. staged learning and sensitivity to hyperparameters
>
> We thank the reviewer for raising this point. In our setting, the "fast"
> module learns a set of cross-session motifs, and the "slow" module
> captures how these motifs are expressed over longer timescales. If one
> first learns the fast dynamics and then learns the slow dynamics on top
> of fixed motifs, this essentially corresponds to optimizing the same
> overall objective with respect to two disjoint parameter blocks in
> sequence (first fast, then slow), rather than jointly. Importantly, this
> two-stage view only makes sense when the motifs are held fixed in the
> second stage; if the motifs are allowed to keep changing while
> re-estimating the slow dynamics, the procedure effectively collapses
> back to a single joint optimization, and the notion of "learning slow
> dynamics of already-learned fast components" becomes ill-defined.
>
> Empirically, we have compared this staged procedure to the joint
> training in the experiment of BCM. Once the motifs have converged, the
> two approaches yield very similar results. To make this explicit, in the
> revised appendix we now include the evolution of the predicted motif
> scales across sessions for both training schemes (see Appendix A.6 and
> Fig. 8). Joint training provides a slightly better fit to the directly
> inferred motif scale curves, but the overall behavior is consistent.
>
> | Models | $\lVert W \rVert_2$ alignment (across-session) | EV (within-session) | Correlation between $\mathbf{c}$ and $\mathbf{c}\_{pred}$ |
> |---|---:|---:|---:|
> | STEER (jointly learning) | 0.9606 ± 0.0036 | 0.9397 ± 0.0008 | 0.9934 ± 0.0016 |
> | STEER (staged learning) | 0.9602 ± 0.0076 | 0.9310 ± 0.0022 | 0.9866 ± 0.0094 |
>
> *Table: Model performance by different training procedures in BCM synthetic dataset.*
>
> Regarding the hyperparameter $\lambda_{slow}$: in the revised version we
> also report a sensitivity analysis in the appendix. We sweep
> $\lambda_{slow}$ over a broad, reasonable range and find that both
> performance and the qualitative conclusions are stable across this
> range, indicating that the method is not particularly sensitive to the
> exact choice of $\lambda_{slow}$.
>
> # Q2: Definition of $\Delta W$ in Fig. 4d
>
> Sorry for the confusion. We've already added the definition of
> $\Delta W$ in the figure (Appendix Fig. 2.)
>
> We again thank the reviewer for the insightful comments. You have
> prompted us to sharpen our claims around identifiability, to provide
> additional ablations on training schemes and regularization, and to
> clarify how $\Delta W$ is defined and interpreted in the figures. We
> believe these changes substantially strengthen the conceptual clarity
> and empirical support for STEER.

---

### Official Review · Reviewer_B3ZH · 2025-11-01

**Soundness:** 2
**Presentation:** 3
**Contribution:** 2
**Rating:** 4
**Confidence:** 4

**Summary:**

This paper introduces STEER, a dual-timescale recurrent framework for inferring long-term neural plasticity rules from longitudinal stimulation data. STEER models fast neural activity within sessions using a low-rank recurrent network, while slow structural changes across sessions are captured by a learnable latent dynamical system driven by stimulation input. Experiments on synthetic benchmarks and Parkinson’s DBS data show that STEER allows disentangle short- and long-term dynamics, recovers biologically interpretable motifs, and generalizes to unseen stimulation protocols.

**Strengths:**

1. **Motivation:** The paper addresses an underexplored area by extending short-term plasticity modeling toward the longer timescales of circuit reorganization. The proposed framework offers a structured, data-driven way to describe how stimulation may gradually reshape network connectivity.
2. **References:** The related research is carefully reviewed, linking established neuroscience findings on Hebbian and homeostatic mechanisms with recent machine learning approaches for recurrent dynamics and meta-learning. This grounding strengthens the biological and methodological motivation.
3. **Evaluation:** The method is tested on two synthetic datasets (Lorenz and BCM) and one real Parkinson’s DBS dataset. The experiments provide supporting evidence that the model can capture slow network adaptations and generate interpretable patterns, though further validation would be beneficial.

**Weaknesses:**

1. **Evaluation:** Results on the BCM and Parkinson’s DBS datasets appear modest, and there is a visible mismatch between the ground truth in Fig. 4(a) and the model output in Fig. 4(b), suggesting partial recovery of the connection change.

2. **Baselines:** Only one baseline (MD-SSM) is evaluated on the BCM simulation. Including other relevant machine learning approaches discussed in the related work would strengthen the empirical comparison.

3. **Benchmark:** The evaluation spans two synthetic and one real dataset. Additional simulations or real neural datasets would better demonstrate the method’s generalization and robustness across experimental settings.

**Questions:**

1. Could the authors discuss how choices such as rank, learning rate, or regularization impact performance? What might explain the prediction gap between the ground truth in Fig. 4(a) and the inferred results in Fig. 4(b)?

2. How does the method scale to larger networks or longer time series? Please comment on computational cost and potential limitations for larger-scale simulations.

3. Since the model infers latent dynamics and connectivity from observed activity, how does it avoid learning spurious correlations due to partial observations and noises?

4. More implementation details (e.g., initialization, optimizer settings, runtime) would be helpful for reproducibility.

---

> ### Author Response · Authors · 2025-11-24
> **Response to Reviewer B3ZH (1/8)**
>
> We would like to thank the reviewer for the careful and constructive
> feedback, and for recognizing the motivation, grounding in prior work,
> and breadth of datasets. Below, we address the concerns point-by-point
> and have incorporated the corresponding clarifications and analyses in
> the revised version.
>
> # W1: Explaining the prediction gap between ground-truth and inferred connectivity.
>
> Thanks for pointing this out. Our goal in the BCM experiment is not to
> reconstruct every individual recurrent weight, but to recover the
> plasticity rule in a low-dimensional latent space and the resulting
> *pattern* of connection changes across sessions. A visible mismatch
> between the ground truth and the inferred results in BCM experiments
> therefore does not contradict our main objective, for several reasons:
>
> -   **Low-rank approximation and identifiability.** In the BCM
>     benchmark, the true recurrent weight matrix is not constrained to be
>     low rank, whereas our model represents $W\_k$ via a low-rank motif
>     decomposition. This architectural mismatch inevitably introduces
>     approximation error at the level of individual entries. Moreover,
>     element-wise equality of $W\_k$ is in general not identifiable under
>     low-rank constraints and shared input/readout, so what we hope to
>     recover reliably is the *trajectory and structure* of connection
>     changes that matter for the dynamics, rather than exact entry-wise
>     weights.
>
> To address the concern about "partial recovery", in the revised
> manuscript we have made the following changes:
>
> -   Figure 4 has been updated to include a direct quantitative
>     comparison of the *evolution of the recurrent weights* across
>     sessions. In Fig. 4(c), we now show that the session-by-session
>     trajectory of $\\| W \\|_2$ inferred by STEER closely tracks the
>     ground truth. The right panel of Fig. 4(c) reports the correlation
>     between the true and inferred $\\| W \\|_2$ trajectories,
>     demonstrating that STEER recovers the *time course of connection
>     changes*.
>
> -   In Appendix Fig. 2, we further visualize $W_k$ and $\Delta W$ at
>     multiple sessions (1, 500, 1000) for the ground truth, STEER, and
>     baselines. While some local discrepancies are expected due to the
>     low-rank parameterization, the global structure and sign pattern of
>     the connectivity changes are well preserved in STEER and clearly
>     differ from the baselines. This supports that the model captures the
>     *trajectory and structure* of stimulus-evoked plasticity rather than
>     only performing a modest fit.
>
> -   Fig. 4(e) shows that the inferred motif coefficients $c_k$ are
>     significantly correlated with the principal components of the true
>     sliding thresholds $\theta_k$, i.e., the latent variables that drive
>     the BCM rule. Thus, even when the visual match between individual
>     weights is not perfect, the learned slow embedding is strongly
>     aligned with the true plasticity mechanism.
>
> -   We also adapted a recently proposed *full-rank* framework (the hierarchical PLRNN baseline) that
>     allows session-by-session variability in $W_k$ and compared it to
>     our low-rank model. In the setting
>     without slow-dynamics forecasting, the full-rank and low-rank models
>     achieve comparable performance in recovering the coarse trajectory
>     of connectivity changes (see Fig.4). At the same time, a growing
>     body of theoretical and experimental work suggests that learning-
>     and context-dependent changes in population activity and effective
>     connectivity are often well captured by low-dimensional,
>     approximately low-rank structure. Our parametrization therefore
>     provides a more computationally efficient and neuroscience-motivated
>     inductive bias for modeling plasticity.
>
> Taken together, these additions clarify that STEER recovers the
> *temporal evolution and low-dimensional structure* of the connection
> changes in the BCM benchmark, which is the quantity our framework is
> designed to identify, rather than a perfect reconstruction of the
> recurrent weights.

---

> ### Author Response · Authors · 2025-11-24
> **Response to Reviewer B3ZH (2/8)**
>
> # W2: Lack of baselines
>
> We thank the reviewer B3ZH for the suggestion. We have added two baselines, including `MD-SSM` and `hierarchical PLRNN`.
>
> *The choice of baselines on the BCM simulation* was motivated by the modelling assumptions of existing machine-learning approaches. Most of these methods (e.g., `MD-SSM`) are built around a session-specific latent embedding that is treated as independent across sessions, without modelling (i) the temporal evolution of these embeddings or (ii) the structured relationships between recurrent weights across sessions. In that sense, MD-SSM is representative of a broad class of approaches that can capture between-session variability but do not explicitly model slow plasticity dynamics.
>
> We have additionally implemented a `hierarchical PLRNN` baseline. The original hierarchical PLRNN formulation introduces a hierarchical prior over parameters but does not learn session-specific embeddings from data in a way that can track session-to-session plasticity trajectories, and thus cannot directly generalize to our setting. We therefore adapted the model by
> introducing learned session-specific latent variables as the hierarchical prior that links these across sessions. This modification allows hierarchical PLRNN to serve as a genuinely competitive baseline for our task, rather than a mis-specified model. Typically, a hierarchical PLRNN is a full-rank dynamical system. However, we argue that this full-rank structure makes it less suitable for capturing the low-rank nature of neural dynamics.
> | Models | Predictive performance (EV ↑) | Dynamical similarity (DSA ↓) |
> |---|---:|---:|
> | Low-rank MD-SSM (rank = 3) | 0.4891±0.1004 | 0.1700±0.0001 |
> | Full-rank MD-SSM | 0.7774±0.0646 | 0.1598±0.0176 |
> | Hierarchical-PLRNN (full rank) | 0.9313±0.0569 | 0.1519±0.0245 |
> | **STEER without forecasting (rank = 3)** | **0.9575±0.0236** | 0.1694±0.0001 |
> | **STEER forecasting all sessions (rank = 3)** | 0.9471±0.0165 | **0.1301±0.0483** |
>
> *Table: Model performance across random seeds in Lorenz synthetic dataset.*
>
>
> | Models | $\lVert W \rVert \_2$ alignment (across-session) | EV (within-session) |
> |---|---:|---:|
> | Low-rank MD-SSM without forecasting (rank 2) | 0.1897±0.2250 | 0.9348±0.0111 |
> | Full-rank MD-SSM without forecasting| 0.0053±0.1932 | 0.9502±0.0048 |
> | Hierarchical-PLRNN without forecasting (full-rank) | 0.9743±0.0041 | 0.9479±0.0013 |
> | **STEER without forecasting (rank 2)** | 0.9706±0.0038 | 0.9460±0.0012 |
> | **STEER with forecasting 5 sessions (rank 2)** | 0.9606±0.0036 | 0.9397±0.0008 |
>
> *Table: Model performance across random seeds in BCM synthetic dataset.*

---

> ### Author Response · Authors · 2025-11-24
> **Response to Reviewer B3ZH (3/8)**
>
> # W3: Consideration of benchmark design
>
> We fully agree that, in principle, more datasets are always desirable. Our current choice of two synthetic and one real dataset was, however, deliberate: it is designed as a progression from fully controlled ground-truth settings to a challenging real neural experiment, while covering several qualitatively different experimental regimes.
>
> 1.  **Lorenz system with parameter evolution (synthetic, fully
>     controlled).** We first evaluate STEER using the Lorenz system, a
>     chaotic dynamical system where \"plasticity\" is modeled as a slow,
>     explicitly specified evolution of the ODE parameters. This setup
>     provides complete ground truth for both fast and slow variables,
>     enabling us to assess whether STEER can correctly recover the slow
>     plasticity rule and generalize across distinct dynamical regimes and
>     unseen parameter trajectories. The synthetic plasticity rule is
>     designed to reflect biological neural networks, where synaptic
>     connections are strengthened or weakened based on various factors.
>     Specifically, we model two parameters increasing while one decays,
>     mimicking the balance between potentiation and depression in neural
>     connections. Meanwhile, the Lorenz system serves as a benchmark for
>     dynamical system reconstruction, offering a simple but effective
>     environment for testing our approach. This controlled scenario
>     allows us to evaluate the core dynamics of our model, providing a
>     solid foundation for future extensions to more complex, biologically
>     relevant benchmarks involving intricate plasticity mechanisms and
>     neural network behavior.
>
> 2.  **BCM-based recurrent network (synthetic, canonical Hebbian
>     plasticity with latent ground truth).** The second benchmark is a
>     recurrent neural population model equipped with a classical BCM
>     plasticity rule. BCM is a canonical Hebbian-type synaptic plasticity
>     rule that is widely used as a textbook model of activity-dependent
>     learning. Crucially for our purposes, BCM includes a sliding
>     modification threshold, which provides an explicit ground-truth
>     latent plasticity variable. This dataset is therefore not just
>     "another simulation": it lets us quantitatively evaluate whether
>     STEER can (i) recover the structure of a Hebbian plasticity rule at
>     the population level and (ii) accurately track the hidden sliding
>     threshold over time. In other words, it bridges the gap between
>     abstract dynamical systems (Lorenz) and biophysically motivated
>     synaptic plasticity, while still giving us a well-defined target for
>     assessing inference quality.
>
> 3.  **PD--DBS longitudinal rat dataset (real neural data under
>     intervention).** Finally, we apply STEER to a real
>     electrophysiological dataset with longitudinal recordings under
>     Parkinsonian lesion and deep brain stimulation. This setting
>     stresses the method under realistic experimental variability and
>     intervention, and we show that the inferred slow variables align
>     with changes in functional connectivity and behavioral improvement.
>
> Together, these three benchmarks address (i) abstract versus
> biophysically motivated plasticity rules, and (ii) fully known versus
> unknown ground truth. We believe this selection offers a comprehensive
> and balanced evaluation, providing strong insight into the
> generalization and robustness of STEER across diverse experimental
> settings.
>
> We recognize that understanding the dynamics of plasticity, particularly
> plasticity induced by long-term stimulation, is a key challenge in
> neuroscience. This is especially relevant for therapeutic interventions
> such as electrical stimulation, which are increasingly used to treat
> neural system diseases. However, there is a lack of publicly available
> datasets specifically addressing this issue, which hinders progress in
> the field. Through this work, we aim to contribute a robust framework
> that can help address this gap, providing insights into how long-term
> stimulation affects neural plasticity. We believe this is a crucial step
> toward advancing our understanding of neural system diseases and their
> treatment, and we hope our approach will inspire further research, the
> development of new datasets, and a deeper exploration of plasticity in
> therapeutic contexts.

---

> ### Author Response · Authors · 2025-11-24
> **Response to Reviewer B3ZH (4/8)**
>
> # Q1: Hyperparameters evaluation
>
> In the revised version we have added a evaluation of the impact of rank,
> learning rate, and regularization (new Appendix A.7). Overall, the
> evaluation show that our model performance is robust to these choices.
>
> # Q2: Scalability and computational cost.
>
> Our method is designed to scale to larger networks and longer time
> series by explicitly exploiting low-rank structure and a low-dimensional
> slow law.
>
> On the *network size* side, a naïve formulation that stores a separate
> recurrent matrix $W_k \in \mathbb{R}^{N \times N}$ for each of the $D$
> sessions would require $O(DN^2)$ parameters and $O(N^2)$ cost per time
> step. In STEER, we instead factorize the across-session connectivity
> tensor with a rank-$R$ CP decomposition and session-dependent motif
> coefficients $c_k$, so the number of parameters scales as $O(NR + DR)$
> rather than $O(DN^2)$. Applying the recurrent connectivity within the
> RNN only requires $O(NR)$ operations per time step (for fixed rank $R$),
> since the update is implemented via a sum of rank-one motifs
> $a_r b_r^\top$ weighted by $c_{k,r}$ (Eq. (4)). In practice we use small
> ranks, so increasing the number of recorded units from tens to hundreds
> increases compute roughly linearly, and memory is dominated by the
> shared motif factors $A,B$ plus the session coefficients $c_k$, not by
> storing full $N \times N$ matrices.
>
> On the *temporal* side, the fast within-session dynamics is a standard
> RNN-like integration (Eq. (4)), so the cost scales linearly with the
> sequence length $T$: $O(D T N R)$ for forward passes, with
> backpropagation through time adding the usual constant factor. We
> already handle Lorenz trajectories with $T = 250$ time steps and
> $D = 100$ parameter-evolving systems, and BCM trajectories with
> $T = 100$ time steps and $D = 1000$ sessions.
>
> The *slow plasticity law* operates in a low-dimensional embedding
> $z_k \in \mathbb{R}^P$ with a small residual network $g_\theta$, and the
> readout from $z_k$ to motif scales $c_k$ is linear/sparse (Eq. (8)), so
> its cost is $O(DP^2 + DRP)$ and negligible compared to the fast RNN when
> $P,R \ll N$.
>
> The main limitation is that training still requires backpropagation
> through the fast dynamics and the slow law, so the overall cost is
> linear in both the number of sessions $D$ and the within-session length
> $T$. Very long sequences (e.g., millions of time steps) would require
> truncated BPTT or multi-scale training schemes to keep memory usage
> manageable. Similarly, although the low-rank factorization avoids an
> $N^2$ blow-up, extremely large populations (e.g., $N \gg 10^3$) would
> benefit from additional structure such as block-diagonal or sparse
> motifs.
>
> Overall, because both the recurrent connectivity and the plasticity rule
> live in low-dimensional motif and embedding spaces, STEER scales
> linearly in network size and sequence length for fixed rank, and its
> computational cost is dominated by the same operations as a standard
> low-rank RNN, rather than by storing or updating full $N \times N$
> connectivity for each session.
>
> # Q3: Avoiding spurious correlations under noise and partial observability
>
> In principle, any method that infers latent dynamics from observed
> activity could overfit to noise or partial observations. To reduce
> spurious correlations arising from partial observability and noise,
> STEER incorporates strong inductive biases: (i) a low-rank structure
> which restricts recurrent connectivity to a low-dimensional,
> biologically structured manifold; (ii) smooth latent dynamics and
> regularization on the slow motif scales; and (iii) an explicit
> dependence of the slow law on stimulation input. These constraints bias
> the model toward learning stable, stimulation-locked motifs rather than
> fitting transient idiosyncrasies in the data.
>
> To explicitly test whether our model is "hallucinating" structure that
> is not present in the data, we have added a shuffle control experiment
> in the appendix A.5 of the revised manuscript. Specifically, we destroy the
> temporal structure of the data by shuffling the activity across sessions
> (so that observations no longer follow the original temporal order) and
> then evaluate the model on this shuffled dataset. In this setting, the
> inferred latent trajectories no longer form a smooth, continuously
> evolving curve. Instead, they become fragmented and lack consistent
> temporal trends. This control shows that when the data do not contain
> genuine temporal dependencies, the model does not infer an artificial,
> smoothly varying latent process. The qualitatively smooth latent
> dynamics and structured connectivity we report in the main text
> therefore reflect the temporal structure present in the real data,
> rather than being an artifact of noise or partial observability.
>
> # Q4: Implementation details and reproducibility
>
> The implementation details of our framework have been added in the appendix, including the initialization, optimizer settings, network size and hyperparameters.

---

> ### Author Response · Authors · 2025-11-27
> **Response to Reviewer B3ZH (5/8)**
>
> To make the hyperparameter impact point more concrete, we now add a new
> table that reports an explicit sensitivity study over rank, learning
> rate, hidden dimension, forecasting session and regularization strength.
>
> Across all three datasets, **STEER is robust to hyperparameter
> choices**, exhibiting broad "plateaus" where performance changes only
> mildly.
>
> On **Lorenz**, predictive EV quickly reaches a stable regime as **rank**
> increases (e.g., around rank $\approx 3$ for most hidden dimensions),
> and remains similarly stable across a wide range of **regularization
> strengths** ($\lambda$); the learning rate is also well-tolerated within
> a reasonable range, with degradation mainly at overly small values.
>
> | Hidden dim \ Tensor rank | 1     | 2     | 3     | 4     | 5     | 6     | 7     | 8     | 9     | 10    |
> |--------------------------|-------|-------|-------|-------|-------|-------|-------|-------|-------|-------|
> | 8                        | -1.109| 0.729 | 0.864 | 0.772 | 0.867 | 0.803 | 0.390 | 0.717 | 0.934 | 0.942 |
> | 12                       | -0.619| 0.806 | 0.908 | 0.898 | 0.933 | 0.890 | 0.852 | 0.963 | 0.925 | 0.928 |
> | 16                       | -1.170| 0.750 | **0.986** | 0.937 | 0.959 | 0.977 | 0.920 | 0.936 | 0.966 | 0.950 |
> | 20                       | -1.084| 0.773 | 0.974 | 0.883 | 0.931 | 0.919 | 0.950 | 0.983 | 0.910 | 0.920 |
> | 24                       | -1.134| 0.773 | 0.962 | 0.979 | 0.956 | 0.892 | 0.978 | 0.905 | 0.943 | 0.980 |
> | 28                       | -0.885| 0.799 | 0.964 | 0.965 | 0.840 | 0.947 | 0.861 | 0.973 | 0.896 | 0.947 |
>
> *Table: Predictive performance (EV) across different tensor ranks and hidden dimensions on Lorenz benchmark.*
>
> &nbsp;
>
> | Hidden dim \ Learning rate | 0.01 | 0.005 | 0.001 | 0.0005 | 0.0001 |
> |--------------------------|---:|---:|---:|---:|---:|
> | 12 | 0.878 | 0.896 | 0.761 | 0.697 | -0.009 |
> | 16 | 0.949 | 0.923 | 0.914 | 0.922 | 0.354 |
> | 20 | 0.942 | 0.952 | 0.935 | 0.879 | 0.354 |
> | 24 | 0.963 | **0.972** | 0.929 | 0.882 | 0.730 |
> | 28 | 0.944 | 0.964 | 0.945 | 0.905 | 0.857 |
>
> *Table: Predictive performance (EV) across different learning rates and hidden dimensions at rank=3 on Lorenz benchmark.*
>
> &nbsp;
>
> | Hidden dim \ Learning rate | 0.01 | 0.005 | 0.001 | 0.0005 | 0.0001 |
> |--------------------------|---:|---:|---:|---:|---:|
> | 12 | 0.169 | 0.015 | 0.169 | 0.082 | 0.102 |
> | 16 | 0.167 | 0.169 | 0.169 | 0.167 | 0.169 |
> | 20 | 0.112 | 0.156 | 0.168 | 0.166 | 0.163 |
> | 24 | 0.099 | 0.161 | 0.169 | 0.169 | 0.166 |
> | 28 | 0.152 | 0.169 | 0.169 | 0.148 | 0.132 |
>
> *Table: Dynamical similarity (DSA) across different learning rates and hidden dimensions at rank=3 on Lorenz benchmark.*
>
> &nbsp;
>
> | Hidden dim \ $\lambda$ | 0.1 | 0.2 | 0.3 | 0.4 | 0.5 | 0.6 | 0.7 | 0.8 | 0.9 | 1 | 2 | 5 | 10 |
> |--------------------------|---:|---:|---:|---:|---:|---:|---:|---:|---:|---:|---:|---:|---:|
> | 12 | 0.795 | 0.883 | 0.903 | 0.911 | 0.928 | 0.913 | 0.954 | 0.930 | 0.810 | 0.908 | 0.755 | 0.835 | 0.943 |
> | 16 | 0.932 | 0.941 | 0.915 | 0.970 | 0.967 | 0.926 | 0.902 | 0.829 | 0.965 | 0.980 | 0.980 | 0.935 | 0.936 |
> | 20 | 0.981 | 0.921 | 0.947 | 0.950 | 0.929 | 0.772 | 0.892 | 0.856 | 0.979 | 0.925 | 0.909 | 0.939 | 0.884 |
> | 24 | 0.961 | 0.905 | 0.820 | 0.861 | 0.970 | 0.978 | 0.970 | 0.962 | 0.943 | 0.962 | 0.973 | 0.879 | 0.922 |
> | 28 | 0.948 | 0.963 | 0.981 | 0.874 | 0.955 | 0.953 | 0.976 | 0.742 | 0.950 | 0.964 | 0.955 | 0.913 | 0.946 |
>
> *Table: Predictive performance (EV) across different regularization coefficient $\lambda$ at rank=3 on Lorenz benchmark.*
>
> &nbsp;
>
> | Hidden dim \ $\lambda$ | 0.1 | 0.2 | 0.3 | 0.4 | 0.5 | 0.6 | 0.7 | 0.8 | 0.9 | 1 | 2 | 5 | 10 |
> |--------------------------|---:|---:|---:|---:|---:|---:|---:|---:|---:|---:|---:|---:|---:|
> | 12 | 0.118 | 0.122 | 0.145 | 0.162 | 0.165 | 0.170 | 0.075 | 0.166 | 0.037 | 0.145 | 0.202 | 0.033 | 0.155 |
> | 16 | 0.121 | 0.169 | 0.164 | 0.169 | 0.165 | 0.165 | 0.125 | 0.169 | 0.080 | 0.164 | 0.142 | 0.156 | 0.159 |
> | 20 | 0.169 | 0.169 | 0.169 | 0.151 | 0.152 | 0.158 | 0.166 | 0.051 | 0.038 | 0.169 | 0.145 | 0.143 | 0.103 |
> | 24 | 0.145 | 0.033 | 0.167 | 0.133 | 0.089 | 0.118 | 0.098 | 0.144 | 0.152 | 0.147 | 0.169 | 0.128 | 0.145 |
> | 28 | 0.165 | 0.149 | 0.024 | 0.122 | 0.105 | 0.170 | 0.038 | 0.168 | 0.120 | 0.169 | 0.168 | 0.081 | 0.125 |
>
> *Table: Dynamical similarity (DSA) across different regularization coefficient $\lambda$ at rank=3 on Lorenz benchmark.*

---

> ### Author Response · Authors · 2025-11-27
> **Response to Reviewer B3ZH (6/8)**
>
> On the **BCM** synthetic dataset, both **within-session EV** and
> plasticity-related metrics (e.g., correlation with ground-truth sliding
> thresholds and cross-session $\\lVert W \rVert_2\$ alignment) remain consistently
> high across learning rate, rank, and $\lambda\_{\text{slow}}$, with only
> a gradual drop when extrapolating to very large numbers of forecasting
> sessions.
>
> | Metrics \ Learning rate | 0.05 | 0.01 | 0.005 | 0.001 | 0.0005 |
> |--------------------------|---:|---:|---:|---:|---:|
> | EV | 0.769 | 0.926 | 0.938 | 0.935 | 0.923 |
> | $\\lVert W \rVert_2\$ alignment | -0.0049 | 0.939 | 0.942 | 0.959 | 0.948 |
> | Max correlation | 0.127 | 0.689 | 0.702 | 0.728 | 0.716 |
>
> *Table: Model performance across different learning rates at rank=2 on BCM benchmark.*
>
> &nbsp;
>
> | Metrics \ Tensor rank | 1 | 2 | 3 | 4 | 5 | 6 | 7 | 8 | 9 | 10 |
> |--------------------------|---:|---:|---:|---:|---:|---:|---:|---:|---:|---:|
> | EV | 0.930 | 0.938 | 0.935 | 0.937 | 0.938 | 0.946 | 0.943 | 0.937 | 0.942 | 0.942 |
> | $\\lVert W \rVert_2\$ alignment | 0.977 | 0.942 | 0.946 | 0.957 | 0.961 | 0.970 | 0.961 | 0.978 | 0.969 | 0.980 |
> | Max correlation | 0.730 | 0.702 | 0.713 | 0.712 | 0.719 | 0.731 | 0.716 | 0.743 | 0.745 | 0.738 |
>
> *Table: Model performance across different tensor ranks on BCM benchmark.*
>
> &nbsp;
>
> | Metrics \ $\lambda_{slow}$ | 0.1 | 0.2 | 0.3 | 0.4 | 0.5 | 0.6 | 0.7 | 0.8 | 0.9 | 1 | 2 | 5 | 10 |
> |---|---:|---:|---:|---:|---:|---:|---:|---:|---:|---:|---:|---:|---:|
> | EV | 0.944 | 0.942 | 0.941 | 0.939 | 0.944 | 0.940 | 0.943 | 0.938 | 0.941 | 0.938 | 0.934 | 0.941 | 0.943 |
> | $\\lVert W \rVert_2\$ alignment | 0.950 | 0.964 | 0.960 | 0.948 | 0.954 | 0.954 | 0.974 | 0.935 | 0.945 | 0.942 | 0.960 | 0.979 | 0.981 |
> | Max correlation | 0.708 | 0.723 | 0.716 | 0.716 | 0.718 | 0.715 | 0.734 | 0.689 | 0.709 | 0.702 | 0.722 | 0.731 | 0.741 |
>
> *Table: Model performance across different consistency regularization coefficient $\lambda_{slow}$ at rank=2 on BCM benchmark.*
>
> &nbsp;
>
> | Metrics \ Forecasting session number | 1 | 2 | 3 | 4 | 5 | 6 | 7 | 8 | 9 | 10 |
> |---|---:|---:|---:|---:|---:|---:|---:|---:|---:|---:|
> | EV | 0.938 | 0.936 | 0.939 | 0.939 | 0.938 | 0.940 | 0.940 | 0.936 | 0.944 | 0.940 |
> | $\\lVert W \rVert_2\$ alignment | 0.976 | 0.970 | 0.958 | 0.967 | 0.942 | 0.973 | 0.964 | 0.947 | 0.971 | 0.968 |
> | Max correlation | 0.744 | 0.515 | 0.708 | 0.512 | 0.585 | 0.458 | 0.264 | 0.178 | 0.440 | 0.346 |
>
> &nbsp;
>
> | Metrics \ Forecasting session number | 11 | 12 | 13 | 14 | 15 | 16 | 17 | 18 | 19 | 20 |
> |---|---:|---:|---:|---:|---:|---:|---:|---:|---:|---:|
> | EV | 0.940 | 0.932 | 0.938 | 0.932 | 0.938 | 0.938 | 0.939 | 0.941 | 0.935 | 0.937 |
> | $\\lVert W \rVert_2\$ alignment | 0.978 | 0.759 | 0.962 | 0.765 | 0.821 | 0.649 | 0.214 | 0.149 | 0.496 | 0.321 |
> | Max correlation | 0.724 | 0.720 | 0.738 | 0.728 | 0.702 | 0.732 | 0.729 | 0.700 | 0.738 | 0.723 |
>
> *Table: Model performance across different forecasting session numbers at rank=2 on BCM benchmark.*

---

> ### Author Response · Authors · 2025-11-27
> **Response to Reviewer B3ZH (7/8)**
>
> Finally, on **PD-DBS**, EV varies smoothly with **rank**, **hidden
> dimension**, and $\lambda_{\text{slow}}$, and the similarity between
> inferred coupling changes and observed functional connectivity changes
> remains comparably stable across these hyperparameter sweeps.
>
> | Metrics \ Tensor rank | 5 | 6 | 7 | 8 | 9 | 10 | 12 | 14 |
> |---|---:|---:|---:|---:|---:|---:|---:|---:|
> | EV | 0.896 | 0.927 | 0.924 | 0.944 | 0.943 | 0.921 | 0.960 | 0.977 |
> | $\Delta \mathbf{c}$-$\Delta FC$ alignment | 0.650 | 0.848 | 0.818 | 0.911 | 0.901 | 0.885 | 0.883 | 0.781 |
>
>
> | Metrics \ Tensor rank | 16 | 18 | 20 | 22 | 24 | 32 | 64 | 128 |
> |---|---:|---:|---:|---:|---:|---:|---:|---:|
> | EV | 0.983 | 0.960 | 0.983 | 0.980 | 0.981 | 0.976 | 0.985 | 0.990 |
> | $\Delta \mathbf{c}$-$\Delta FC$ alignment | 0.803 | 0.834 | 0.818 | 0.872 | 0.946 | 0.908 | 0.922 | 0.904 |
>
> *Table: Model performance across different tensor ranks on PD-DBS Rat benchmark.*
>
> &nbsp;
>
> | Metrics \ Hidden dim | 20 | 30 | 60 | 100 | 150 | 200 | 300 |
> |---|---:|---:|---:|---:|---:|---:|---:|
> | EV | 0.812 | 0.906 | 0.929 | 0.942 | 0.972 | 0.984 | 0.990 |
> | $\Delta \mathbf{c}$-$\Delta FC$ alignment | 0.896 | 0.923 | 0.857 | 0.884 | 0.925 | 0.919 | 0.892 |
>
> *Table: Model performance across different hidden dimensions at rank=24 on PD-DBS Rat benchmark.*
>
> &nbsp;
>
> | Metrics \ $\lambda$ | 0.01 | 0.1 | 0.2 | 0.3 | 0.4 | 0.5 | 0.6 | 0.7 | 0.8 | 0.9 | 1 | 2 | 5 | 10 |
> |---|---:|---:|---:|---:|---:|---:|---:|---:|---:|---:|---:|---:|---:|---:|
> | EV | 0.970 | 0.986 | 0.976 | 0.953 | 0.985 | 0.983 | 0.973 | 0.977 | 0.985 | 0.987 | 0.980 | 0.980 | 0.976 | 0.974 |
> | $\Delta \mathbf{c}$-$\Delta FC$ alignment | 0.874 | 0.914 | 0.892 | 0.902 | 0.910 | 0.900 | 0.894 | 0.882 | 0.878 | 0.817 | 0.884 | 0.895 | 0.806 | 0.879 |
>
> *Table: Model performance across different regularization parameters $\lambda$ at rank=24 on PD-DBS Rat benchmark.*
>
> We thank the reviewer again for the thoughtful evaluation. The suggested
> clarifications, additional baselines, and analyses have helped us
> significantly strengthen both the empirical and conceptual presentation
> of STEER.

---

> ### Author Response · Authors · 2025-12-04
> **Response to Reviewer B3ZH (8/8)**
>
> # W3: Additional benchmark
> In response to the lack of benchmarks, we introduce a
> *stimulation-evoked reconfiguration* dataset in which recurrent
> connectivity $\mathbf W(t)$ changes under **closed-loop, adaptively
> optimized** control inputs $\mathbf u(t)$ \[1\]. This benchmark is
> intentionally distinct from our BCM-based synthetic dataset, where both
> the plasticity rule and the input ensemble are fixed and prespecified.
> Here, neural trajectories are induced by an optimized stimulus policy
> that strategically drives the network toward a ring-attractor target
> $\mathbf{W}^{target}$. As a result, this dataset probes whether STEER
> can recover connectivity reconfiguration from population dynamics under
> a qualitatively different regime, where the input distribution and
> resulting dynamics are shaped by optimization rather than by
> hand-designed stimulation.
>
> We have added the full simulation description and results to the revised
> manuscript (see Section 4.3 and Appendix A.1). **Results show that our model outperform baselines in both $W$ inference across sessions and within-session predictive performance.** While we agree that
> evaluating on additional real neural datasets would be valuable, access
> to datasets with sufficiently detailed stimulation metadata and
> ground-truth connectivity proxies is limited; we therefore view the
> newly added closed-loop stimulation benchmark as a complementary and
> challenging test of robustness across experimental regimes, and we will
> pursue further multi-dataset real-data validation.
>
> | Models | $W$ similarity ($\uparrow$) (across-session) | EV ($\uparrow$) (within-session) |
> |---|---:|---:|
> | Low-rank MD-SSM without forecasting (rank 7) | 0.4623 $\pm$ 0.0422 | 0.8866 $\pm$ 0.0297 |
> | Full-rank MD-SSM without forecasting | 0.4733 $\pm$ 0.0579 | 0.7333 $\pm$ 0.0198 |
> | Hierarchical-PLRNN without forecasting (full-rank) | 0.5639 $\pm$ 0.0181 | 0.9706 $\pm$ 0.0009 |
> | **STEER without forecasting (rank 7)** | 0.5222 $\pm$ 0.0650 | 0.9783 $\pm$ 0.0035 |
> | **STEER with forecasting (rank 7)** | **0.6973 $\pm$ 0.0815** | **0.9811 $\pm$ 0.0043** |
>
> *Table: Model performance across random seeds in task learning synthetic dataset.*
>
>
>
> ## Refereces
>
> \[1\] Borra, F., Cocco, S., & Monasson, R. (2024). Task learning through
> stimulation-induced plasticity in neural networks. PRX Life, 2(4),
> 043014.

---

### Official Review · Reviewer_TgyK · 2025-11-03

**Soundness:** 2
**Presentation:** 3
**Contribution:** 2
**Rating:** 4
**Confidence:** 4

**Summary:**

This paper introduces STEER, a method to infer the dynamical rules underlying long-term evolution of recurrent connectivity over sessions/hours/days from neural data. The framework learns low-dimensional latent coefficients for the neural dynamics underlying a session, which themselves evolve over slower timescales – thus allowing the inference of plasticity rules explicitly. The framework is trained by jointly optimising for reconstruction fidelity within a session, regularisation for consistency of inferred plasticity rule across sessions, and also regularisation for smoothness. The authors validate their framework on three tasks – two synthetic tasks involving learning a Lorenz system with varying coefficients and synthetic data generated with a specific, known plasticity rule; and one real dataset of neural recordings from rats undergoing DBS treatment for Parkinson's disease. The main claim is that STEER disentangles effectively the within-session dynamics from cross-session dynamics, allowing better extraction and interpretability of the rules/changes underlying cross-session variability.

**Strengths:**

* The framework seems principled in its design and the approach overcomes disadvantages of certain prior meta-learning approaches by not assuming a specific functional form of the underlying learning rule (e.g., not restricted to just variants of Oja's rule).
* The presentation and figures are mostly clear.
* The experimental evaluations span both synthetic tasks and real data, which is important for such works.

**Weaknesses:**

* The use of DSA is good but you only report a single DSA value of 0.63 with no baseline or control/chance value. DSA is a relative metric, as emphasised in the original paper. Thus, it is not possible to know whether a DSA of 0.63 is good without a baseline comparison. Could the authors provide a baseline on shuffled values of the implicit factor, for example, and show that the actual inferred dynamics have a higher DSA score with the true dynamics?
* In Fig. 4f I am not sure you can claim that you perform comparably to MD-SSM without a statistical significance test. The avg. score for STEER seems lower even when considering error bars. What are the error bars over – sessions or seeds? If the result of a stat. test is not significant then that validates the claim of comparable performance. I would also ask that the plot y axis range be restricted so it's clearer what the difference in performance is.
* There is no quantification for Fig. 4d where it is claimed that STEER better recovers the $\Delta \mathbf{W}$ compared to MD-SSM. As far as I'm concerned, the block structure of both the MD-SSM nor STEER weight changes do not particularly resemble the ground truth weight changes. Could the authors comment on this? The value scales are also quite different and while I understand that exact values/scale may not matter, maybe they need to be normalised so the plots can be compared better?
* There is no comparison against the meta-learning approaches mentioned in the introduction. While I can understand the difference between these methods, I think it is important to see clear evidence to back up the claim that they cannot model the effects of DBS, for example, and also that STEER can recover plasticity rules as those methods do in short-term settings.
* Minor nit: in lines 130-131 you cite Bredenberg et al. and Kepple et al. as fitting synaptic plasticity rules through gradient-based optimisation on observational data. While Bredenberg et al. does do this, to my knowledge, they do not do it on real data. Meanwhile, Kepple et al. do not fit parameters for learning rules – they mainly evaluate different learning rules and curricula in an in silico setting on the basis of learning speeds (and classifying between these on the basis of such observations), again not working with real data to my knowledge.

**Questions:**

* Minor typo in Fig. 1a: "Brian" -> "Brain".
* Why is the evolution of coefficients jagged and not smooth in the Lorenz system plots?
* What is the difference between Fig. 4 and App. Fig. 1? Why are the results so different qualitatively?
* Could the authors provide additional information/motivation on the consistency and smoothness terms? How are the coefficient $\lambda$s tuned?

---

> ### Author Response · Authors · 2025-11-24
> **Response to Reviewer TgyK (1/4)**
>
> We would like to thank the reviewer for the careful and thoughtful assessment of our work and for the constructive suggestions. Below, we address the main concerns point-by-point and have incorporated the corresponding changes in the paper.
>
> # W1: DSA evaluation and missing baseline.
>
> We agree that DSA should be interpreted as a relative measure. In the revised version of Fig.3 and Appendix Fig.3 and 4, we now include explicit baseline comparisons and the chance value on the session-wise shuffled data. The results indicated that our framework is better than other baseline models on both predictive performance and dynamical similarity analysis (lower DSA value means better similarity) on the ground truth plasticity rule.
> | Models | Predictive performance (EV ↑) | Dynamical similarity (DSA ↓) |
> |---|---:|---:|
> | Low-rank MD-SSM (rank = 3) | 0.4891±0.1004 | 0.1700±0.0001 |
> | Full-rank MD-SSM | 0.7774±0.0646 | 0.1598±0.0176 |
> | Hierarchical-PLRNN (full rank) | 0.9313±0.0569 | 0.1519±0.0245 |
> | **STEER without forecasting (rank = 3)** | **0.9575±0.0236** | 0.1694±0.0001 |
> | **STEER forecasting all sessions (rank = 3)** | 0.9471±0.0165 | **0.1301±0.0483** |
> *Table: Model performance across random seeds in Lorenz synthetic dataset.*
>
> # W2: On the comparison to MD-SSM in Fig. 4.
>
> We have revised the figure and the accompanying text to clarify three points: (i) what the error bars represent, (ii) how we assess statistical significance, and (iii) what we mean by "comparable" performance.
>
> **Error bars.** In the revised manuscript, we now explicitly state that the error bars correspond to the mean $\pm$ standard error across random seeds.
>
> **Statistical comparison and parameter-matched baseline.** Structurally, `MD-SSM` represents each session-specific transition as $$W^{(s)}\_{Full-rank\ MD-SSM} = U^{(s)} V^{(s)\top} + W\_{hh},$$
> where the session embedding is used to generate the low-rank factors $U^{(s)}, V^{(s)}$, and $W_{hh} \in \mathbb{R}^{N \times N}$ is a learned, *session-invariant* recurrent matrix.
> In contrast, STEER does not include an additional global $W_{hh}$ and uses the session embedding to infer the coefficients of a low-rank *tensor*, while the basis shared across sessions is fixed.
> As a result, `MD-SSM` has at least $N^2$ more learnable parameters than `STEER` due to $W_{hh}$ alone.
>
> To make the comparison of architectures fair, we have added an ablation in which we remove $W_{hh}$ from MD-SSM,
> $$W^{(s)}_{Low-rank\ MD-SSM} = U^{(s)} V^{(s)\top},$$
> so that the total parameter count is closely matched to that of STEER.
>
> We now report in the revised manuscript that, under this parameter-matched setting, STEER and Low-rank MD-SSM exhibit no statistically significant difference in short-term prediction performance. This quantitative analysis supports our claim that *STEER performs comparably to MD-SSM* when the parameter budgets are aligned.
>
> For completeness, we also retain the original MD-SSM variant with $W_{hh}$, which has a strictly larger parameter count. As expected, this higher-capacity model achieves slightly better short-horizon prediction scores than STEER. We now make this trade-off between overall prediction accuracy, parameter count, and the fidelity of session-to-session variability modeling explicit in the text.
>
> **Visualization (y-axis range).** We admit that the original y-axis range in Fig. 4 made it difficult to see the small differences between methods. We have modified Fig. 4. The y-axis is shown within the relevant performance range. In this way, the relative performance of STEER and the MD-SSM variants is visually clearer. The caption has been updated accordingly.

---

> ### Author Response · Authors · 2025-11-24
> **Response to Reviewer TgyK (2/4)**
>
> # W3: On the quantification and visualization of weight changes.
>
> We agree that a quantitative assessment and a more careful treatment of value scales are necessary to substantiate the claim in the original version of Fig. 4d.
>
> **Quantification of session-wise weight changes.** In the revision, we now explicitly quantify how well each model recovers the ground-truth evolution of the weights across sessions. For each session $s$, we compute the L2-norm of the ground-truth weight matrix $W$ and of the corresponding inferred weights for STEER and baselines. We then perform a min-max normalization of these norms across sessions and compute the correlation between the resulting trajectories and the ground-truth trajectory. Across random seeds, *STEER consistently achieves a substantially higher correlation with the true trajectory than MD-SSM*, indicating that STEER more faithfully captures how $W$ varies from session to session. These correlations are now reported in the revised manuscript.
> | Models | $\lVert W \rVert_2$ alignment (across-session) | EV (within-session) |
> |---|---:|---:|
> | Low-rank MD-SSM without forecasting (rank 2) | 0.1897±0.2250 | 0.9348±0.0111 |
> | Full-rank MD-SSM without forecasting| 0.0053±0.1932 | 0.9502±0.0048 |
> | Hierarchical-PLRNN without forecasting (full-rank) | 0.9743±0.0041 | 0.9479±0.0013 |
> | **STEER without forecasting (rank 2)** | 0.9706±0.0038 | 0.9460±0.0012 |
> | **STEER with forecasting 5 sessions (rank 2)** | 0.9606±0.0036 | 0.9397±0.0008 |
>
> *Table: Model performance across random seeds in BCM synthetic dataset.*
>
> To address the reviewer's concern about value scales and visual comparability, we have (i) normalized the weight matrices $W$ for the ground truth, MD-SSM, and other baselines, and (ii) recomputed and visualized the corresponding $\Delta W$ based on these normalized weights. The updated plots are provided in the revised Appendix Fig. 2, which makes the relative changes and block structure more directly comparable across methods.
>
> **Interpretation of latent low-rank weights.** The true recurrent weight matrix is not constrained to be low rank, whereas our model represents $W_k$ via a low-rank motif decomposition. This architectural mismatch inevitably introduces approximation error at the level of individual entries. Moreover, element-wise equality of $W_k$ is in general not identifiable under low-rank constraints and shared input/readout, so what we can hope to recover reliably is the *trajectory and structure*
> of connection changes that matter for the dynamics, rather than exact entry-wise weights.
>
> In this synthetic setting, STEER typically recovers a latent decomposition into a small number of interacting components that induces a *coarse* modular organization of the dynamics, which is similar to the ground truth. In line with this, we now present Fig. 4d as a qualitative illustration of the inferred latent connectivity across-session evolution, rather than as a claim of exact recovery of the ground-truth weight matrix. Our quantitative evaluation focuses on invariants of the latent dynamics (the trajectory of effective interaction strengths across sessions) that are meaningful under these identifiability constraints.

---

> ### Author Response · Authors · 2025-11-24
> **Response to Reviewer TgyK (3/4)**
>
> # W4: Comparison to meta-learning approaches
>
> We appreciate this concern and agree that the connection should be clearer.
>
> 1.  **Scope of our claim and conceptual difference.** Our intention was *not* to claim that meta-learning approaches are fundamentally unable to model DBS, but rather that they operate at a different modeling level and with different assumptions.
>
> Classical meta-learning work infers *explicit, parametric synaptic update rules*, e.g., algebraic forms of $$\Delta w = f(pre, post, r, \dots),$$ from short-term tasks with access to pre/post activity and often trial-wise reward or error signals. In contrast, STEER is designed as a *data-driven framework*:
>
> - It learns a *latent dynamical law* on the space of recurrent motifs and their scaling coefficients $c_k$.
>
> - This law is *conditioned on stimulation protocols and session-level covariates*.
>
> - It does *not* require specifying an algebraic plasticity rule *a priori*.
>
> In other words, instead of restricting ourselves to a particular functional form of plasticity, we use data to infer how DBS protocols reshape latent motifs over days-weeks.
>
> 2.  **Meta-Learning based black-box models for learning dynamical systems.** In addition to the approach we described, we also introduced other meta-learning-based black-box models as baselines for comparison, such as MD-SSM and Hierarchical PLRNN. MD-SSM attempts to infer session-wise embeddings and then determines the session-specific parameters of the dynamical system. Hierarchical PLRNN, on the other hand, models a full-rank dynamical system where its parameters are inferred by a neural network. These models are primarily designed to infer session-wise dynamical systems, but they fail to capture the evolving session-wise dynamics. Through further comparison, we found that these models underperform relative to our method. Specifically:
> - `MD-SSM` underperforms relative to our method, particularly in capturing the intricate dynamics and long-term trends present in the data.
> -  `Hierarchical PLRNN`, while performing similarly to our approach in terms of overall predictive accuracy, differs fundamentally in terms of model complexity. Hierarchical PLRNN represents a special case of our framework with a full-rank structure, but it requires significantly more parameters. The original hierarchical PLRNN places a hierarchical prior on parameters but lacks learned session-specific embeddings to track plasticity across sessions, so it cannot directly handle our setting. We therefore add learned session-specific latent variables as the hierarchical prior, making hierarchical PLRNN a well-specified and competitive baseline for our task.
>
> - The low-rank assumption reflects the concept that neural systems often operate in lower-dimensional subspaces, where the dynamics can be captured more efficiently. In neuroscience, this is akin to the concept that brain networks typically exhibit latent structure, with plasticity and neural activity evolving along a smaller number of dominant modes rather than in a fully independent manner. By incorporating this assumption, our approach aligns better with the biological constraints of neural systems, offering a more efficient and interpretable model.
>
> # W5: Clarification on citations of prior work (Bredenberg et al., Kepple et al.)
>
> In Sec. 2.2, our intention was not to claim that Bredenberg et al. (2020) and Kepple et al. (2022) fit synaptic plasticity rules directly to real neural data. Bredenberg et al. derive a reward-modulated Hebbian update rule that enables the learning of efficient, task-specific representations under resource and noise constraints. Kepple et al. investigated how curriculum learning can uncover the fundamental learning principles (like target-based and representational learning) that guide
> how tasks are learned, drawing a parallel to biological systems that rely on plasticity mechanisms to adapt to new experiences.
>
> We revised the text to make this distinction explicit. The revised version is "A parallel line of work related to plasticity rules
> inference and derivation. Bredenberg et al. (2020) derived a task-dependent Hebbian plasticity rule to enable the learning of
> efficient, task-specific representations under resource and noise constraints. Kepple et al. (2022) explore how curriculum learning can be used to uncover the learning principles underlying both artificial neural networks and biological brains. ")
>
> **References:**
>
> Kepple, D. R., Engelken, R., & Rajan, K. (2022, January). Curriculum learning as a tool to uncover learning principles in the brain. In International Conference on Learning Representations.
>
> Bredenberg, C., Simoncelli, E., & Savin, C. (2020). Learning efficient task-dependent representations with synaptic plasticity. Advances in neural information processing systems, 33, 15714-15724.

---

> ### Author Response · Authors · 2025-11-24
> **Response to Reviewer TgyK (4/4)**
>
> # Responses to specific questions
>
> -   **Q1:** Typo in Fig. 1a ("Brian" → "Brain"). Thanks for pointing out
>     the typo. We have corrected this typo.
>
> -   **Q2:** Why are Lorenz coefficients jagged rather than smooth?
>     Thanks for pointing out the jagged issues in Fig.3. A longer
>     prediction capability of the plasticity predictor in our framework
>     have addressed this. The updated Fig.3 is shown in the updated
>     version.
>
> -   **Q3:** Difference between Fig. 4 and App. Fig. 1. We apologize for
>     the confusion. In the revised version, we have added a description
>     of these weights and clarified how they are used to produce each
>     figure. We have also updated the corresponding visualization and
>     moved the new version to the appendix to avoid confusion.
>
> -   **Q4:** Consistency and smoothness terms and tuning of coefficients.
>
>     Thanks for pointing this out.
>
>     The **consistency term** is the loss function that minimizes the
>     prediction error of the plasticity embedding across sessions. By
>     minimizing this term, we ensure that the model not only learns
>     session-specific dynamics but also the evolving parameters that
>     govern the long-term plasticity changes across sessions. This term
>     thus helps the model generalize across sessions while maintaining a
>     coherent view of the underlying plasticity rules. Additionally, in
>     the Appendix Fig.7, we evaluate the impact of these hyperparameters
>     on the performance at both the fast-time-scale and slow-time-scale,
>     demonstrating how they influence the model's ability to capture
>     short-term fluctuations and long-term trends in the data.
>
>     The **smoothness term** is introduced to limit the differences
>     between session-wise scaling factors. This term reflects the fact
>     that plasticity induced by long-term stimulation is not expected to
>     change suddenly between sessions; rather, it evolves slowly over
>     time. This smoothness assumption is grounded in the idea that the
>     brain's response to stimulation is gradual, driven by the cumulative
>     effects of ongoing activity and stimulation. The smoothness term
>     helps regularize the model by ensuring that session-to-session
>     transitions are not overly abrupt, which would be biologically
>     implausible.
>
>     Regarding the tuning of the coefficients for these terms, we mainly
>     choose them by evaluating their impact on the prediction performance
>     and the structure alignment between predicted and actual dynamics.
>     We perform a series of experiments to assess how well the model
>     generalizes across sessions and how accurately it captures the
>     evolving dynamics of the system. The optimal coefficients are
>     selected based on a balance between minimizing prediction error and
>     maintaining smooth, consistent transitions between sessions.
>
> We hope these clarifications and additional analyses address your
> concerns. We appreciate the reviewer's careful feedback, which has
> helped us substantially strengthen the empirical and conceptual
> presentation of STEER.

---

### Author Response · Authors · 2025-11-27
**Global Response to Reviewers**

We thank the reviewers for their careful reading and constructive
feedback. In the revision, we addressed four recurring themes: (i) more
direct and metric-faithful evaluation of *slow plasticity inference*,
(ii) fairer and stronger baselines, (iii) clearer discussion of
*identifiability/interpretation* of recovered connectivity and timescale
separation, and (iv) improved presentation and reproducibility.

## 1) Stronger evaluation of the inferred slow plasticity (beyond state prediction).

To more directly assess cross-session plasticity, we expanded our
evaluation with explicit controls and plasticity-focused metrics: (i)
comparisons for DSA on Lorenz, (ii) $\\| W \\|_2$-alignment to quantify
plasticity-related evolution and max correlation with ground-truth
sliding threshold PC metrics in the BCM benchmark, (iii)
$\Delta \mathbf{c}$-$\Delta FC$ alignment on real data, and (iv)
evaluation on session-shuffled data of all dataset, testing whether
inferred slow dynamics persist when temporal structure is destroyed.
Results show that our model captures genuine cross-session slow
dynamics: plasticity-related metrics of data in original session order
significantly exceed the session-shuffled null distribution (t-test, $p<0.0001$
for all datasets).

## 2) Baselines (MD-SSM, hierarchical PLRNN).

We strengthened baselines in two ways. First, we added an adapted
full-rank *hierarchical PLRNN*\[1\] baseline that supports
session-specific latent embeddings for the cross-session setting,
enabling a clearer expressivity--efficiency comparison against our
low-rank motif parameterization. Notably, the hierarchical PLRNN can
also be viewed as a form of meta-learning, where higher-level parameters
capture structure across sessions while lower-level (session-specific)
latent embeddings adapt the model to each individual session. Second, we
expanded MD-SSM results to include both low-rank and full-rank variants,
making capacity trade-offs explicit and clarifying that session-level
forecasting can slightly trade off short-horizon prediction, while
STEER remains competitive on within-session prediction and
additionally forecasts future sessions (e.g., all on Lorenz, five on
BCM).

| Models |EV ↑ | Dynamical similarity (DSA ↓) |
|---|---:|---:|
| Low-rank MD-SSM (rank = 3) | 0.4891±0.1004 | 0.1700±0.0001 |
| Full-rank MD-SSM | 0.7774±0.0646 | 0.1598±0.0176 |
| Hierarchical-PLRNN (full rank) | 0.9313±0.0569 | 0.1519±0.0245 |
| **STEER without forecasting (rank = 3)** | **0.9575±0.0236** | 0.1694±0.0001 |
| **STEER forecasting all sessions (rank = 3)** | 0.9471±0.0165 | **0.1301±0.0483** |
*Table: Model performance across random seeds in Lorenz synthetic dataset.*

| Models |$\lVert W \rVert_2$ alignment (across-session) | EV (within-session) |
|---|---:|---:|
| Low-rank MD-SSM without forecasting (rank 2) | 0.1897±0.2250 | 0.9348±0.0111 |
| Full-rank MD-SSM without forecasting| 0.0053±0.1932 | 0.9502±0.0048 |
| Hierarchical-PLRNN without forecasting (full-rank) | 0.9743±0.0041 | 0.9479±0.0013 |
| **STEER without forecasting (rank 2)** | 0.9706±0.0038 | 0.9460±0.0012 |
| **STEER with forecasting 5 sessions (rank 2)** | 0.9606±0.0036 | 0.9397±0.0008 |

*Table: Model performance across random seeds in BCM synthetic dataset.*

## 3) Interpreting recovered connectivity: identifiability and quantification.

We clarified that exact entry-wise recovery of weights is generally not
identifiable under low-rank structure with shared inputs/readouts.
Accordingly, we frame each motif ($a\_rb\_r^\top$) as a latent
*reorganization direction* governing cross-session changes in effective
connectivity ($W^k$). Motifs capture coordinated changes in the
effective dynamics that govern biologically relevant properties in
latent space (e.g., responsiveness to perturbations, persistence of
pathological activity, and the timescale of plastic
accumulation/saturation).

## 4) Presentation and reproducibility.

We improved figure clarity (definitions, error bars, statistical tests,
axis ranges), expanded implementation details, added hyperparameter
sensitivity analyses and ablations (e.g., staged vs. joint training),
and clarified benchmark motivation and metric choices across
Lorenz/BCM/real DBS.

## Summary

Overall, we sharpen the core claim: STEER identifies a
low-dimensional, protocol-conditioned law governing slow reconfiguration
of latent effective connectivity, supported by stronger
plasticity-focused evaluation, fairer baselines, and clearer
presentation.

## References

\[1\] Brenner, M., Weber, E., Koppe, G., & Durstewitz, D. (2025).
Learning interpretable hierarchical dynamical systems models from time
series data. In Proceedings of the Thirteenth International Conference
on Learning Representations.

---

> ### Author Response · Authors · 2025-12-04
>
> For the additional benchmark, we also compared the performance of our model with baselines. **Results show that our model outperform baselines in both $W$ inference across sessions and within-session predictive performance.**
>
> | Models | $W$ similarity ($\uparrow$) (across-session) | EV ($\uparrow$) (within-session) |
> |---|---:|---:|
> | Low-rank MD-SSM without forecasting (rank 7) | 0.4623 $\pm$ 0.0422 | 0.8866 $\pm$ 0.0297 |
> | Full-rank MD-SSM without forecasting | 0.4733 $\pm$ 0.0579 | 0.7333 $\pm$ 0.0198 |
> | Hierarchical-PLRNN without forecasting (full-rank) | 0.5639 $\pm$ 0.0181 | 0.9706 $\pm$ 0.0009 |
> | **STEER without forecasting (rank 7)** | 0.5222 $\pm$ 0.0650 | 0.9783 $\pm$ 0.0035 |
> | **STEER with forecasting (rank 7)** | **0.6973 $\pm$ 0.0815** | **0.9811 $\pm$ 0.0043** |
>
> *Tabel: Model performance across random seeds in task learning synthetic dataset.*

---

### Author Response · Authors · 2025-12-04
**Global Response to Reviewers and Area Chairs (2/2)**

## (C) Two shuffle controls: testing *reliability* and *identifiability*.

(In response to Reviewer TgyK W1, Reviewer B3ZH Q3, Reviewer sTJk W1,
W2) We introduce two complementary shuffle experiments because there are
two distinct failure modes when claiming *plasticity inference*. One is
a **reliability/interpretability issue**: a high plasticity score could
arise from static session-to-session heterogeneity even if there is *no
coherent slow drift*. The other is an **identifiability issue**: the
model class might *impose* smooth trends through its inductive bias,
making it unclear whether the recovered slow trajectory is truly learned
from the data.

-   Evaluating stage shuffle (**null distribution for our plasticity
    metrics**). We keep the model trained on the true session order
    fixed, then generate many session-order--shuffled data and evaluate
    exactly the same plasticity metrics. Because shuffling breaks
    coherent slow evolution while preserving within-session statistics,
    this provides a direct *chance baseline* for our plasticity
    readouts.

-   Training stage shuffle (ruling out "the model hallucinates drift").
    We also train the same architecture on session-order--shuffled data
    and ask whether it can still reproduce the ordered slow-trend
    patterns. If a similar trend emerged despite the absence of
    chronological structure, it would indicate that the trend is largely
    an artifact of model bias rather than inferred from temporal
    continuity. Instead, **training on shuffled order removes the
    ordered slow-timescale trend**, supporting that our slow plasticity
    dynamics are learned from coherent across-session dynamics present
    only in the correctly ordered data.

## (D) Systematic comparisons with baselines.

(In response to Reviewer TgyK W2, W3, W4, Reviewer B3ZH W2, Reviewer
YUkh Q2) We performed a **systematic comparison** against carefully
chosen baselines (both **meta-learning-based black-box dynamical
models**) that reflect different modeling trade-offs. We report both
**plasticity-inference quality** and **fast-timescale predictive
accuracy** under the same evaluation protocol.

We chose baselines to match the task: modeling session-to-session
variability and evaluating plasticity inference. Importantly, **neither
baseline natively supports slow-timescale forecasting**. Empirically, we
find two key results: (1) MD-SSM variants are competitive for
within-session prediction, but they do not reliably recover **the slow,
session-to-session plasticity evolution**. (2) Additionally, we use an
adapted hierarchical PLRNN baseline for fair comparison. Despite the
baseline's full-rank, session-varying flexibility, our model matches its
performance with **a more structured and efficient parameterization**,
while additionally enabling **explicit slow-timescale plasticity
forecasting**.

## (E) Additional improvements

Reviewers also asked for some clarifications and supplementary details.
In response, we made the following changes:

-   **Presentation:** (In response to Reviewer TgyK W2, Q1, Q2, Reviewer
    YUkh W1) Improved figure clarity and statistical reporting
    throughout, including clearer axis ranges/effect-size presentation
    and explicit error-bar definitions.

-   **Clarification:** (In response to Reviewer TgyK W4, Reviewer YUkh
    W1, W2) We have responded to all clarification-type comments by
    providing point-by-point explanations in our rebuttal and revising
    the corresponding manuscript sections to make the content explicit
    and unambiguous.

-   **Connectivity-change definitions:** (In response to Reviewer TgyK
    Q3, Reviewer sTJk Q2) Tightened the definition and normalization of
    connectivity-change quantities.

-   **Interpretability and robustness:** (In response to Reviewer TgyK
    Q4, Reviewer B3ZH Q1, Reviewer sTJk Q1, Reviewer YUkh Q1) Clarified
    how our structural constraints affect interpretability, analyzed the
    impact of different training procedures on interpretability and
    performance, and added hyperparameter sensitivity experiments to
    assess robustness across settings.

-   **Reproducibility:** (In response to Reviewer B3ZH Q2, Q4) Added
    implementation details and discussions of computational scaling.

Thanks to all the reviewers for your thoughtful and constructive comments. We also sincerely appreciate the Area Chair for carefully reading the discussions and our work. We hope our responses have addressed the concerns clearly, and we look forward to the final assessment.

---

### Author Response · Authors · 2025-12-04
**Global Response to Reviewers and Area Chairs (1/2)**

# Summary of our work.

In neurostimulation, the effects of stimuli accumulate over time as
neural circuits gradually adapt to repeated inputs. However, several
**key questions** remain unresolved: How does network responsiveness
evolve across multiple sessions? When, if ever, does it plateau? And how
do different stimulation protocols shape the adaptation trajectory? Our
research aims to develop **a two-timescale neural dynamics model** in
which a slow-timescale plasticity rule is influenced by stimulus and
fast-timescale neural dynamics. This model goes beyond fitting past
data: it predicts neural-circuit dynamics and implicitly reveals
principles governing long-term evolution, offering clinically relevant
insights into plasticity and adaptation.

Our model separates (i) fast within-session neural dynamics from (ii)
slow cross-session reconfiguration, and links slow changes directly to
stimulation history. Concretely, STEER parameterizes session-specific
recurrent connectivity via a low-rank motif decomposition and models the
session-to-session evolution of motif coefficients with a learnable,
protocol-dependent plasticity rule.

# Rebuttal additions and responses to the reviewers.

In response to reviewer concerns, we added analyses and clarifications
along five main axes.

## (A) Additional benchmark.

(In response to Reviewer B3ZH W3, Q3) We introduce a stimulation-evoked
reconfiguration dataset where recurrent connectivity ($\mathbf W(t)$)
changes under closed-loop, adaptively optimized inputs ($\mathbf u(t)$)
\[1\]. Unlike our BCM-based synthetic dataset with fixed plasticity and
prespecified inputs, here an optimized stimulus policy drives the
network toward a ring-attractor target ($\mathbf W^{target}$). This
benchmark tests whether STEER can recover connectivity reconfiguration
when the input distribution and dynamics are optimization-shaped rather
than hand-designed.

## (B) Quantifying plasticity inference.

(In response to Reviewer TgyK W3, Reviewer B3ZH W1, Reviewer YUkh W3,
Q2) Across datasets, we quantified plasticity inference by checking
whether the model recovers the slow, session-to-session rule/trajectory,
using dataset-appropriate ground truth.

-   **Lorenz (synthetic, known parameter drift)**: Plasticity inference
    is measured by Dynamical Similarity Analysis (DSA) between the
    inferred motif scale slow evolution and the true predefined
    parameter evolution.

-   **BCM network (synthetic, known plasticity):** We quantify
    plasticity inference by how well the model recovers
    session-to-session connectivity drift, measured via alignment
    between inferred and ground-truth $\\| W\\|_2$ trajectories. We also
    validate that inferred motif scales $\mathbf{c}$ reflect homeostatic
    control by correlating $\mathbf{c}$ with PCs of the ground-truth
    sliding threshold $\theta$.

-   **Task learning through stimulation-induced plasticity.** In this
    dataset, we quantify plasticity inference by the correlations
    between the inferred weights and the ground-truth weights across
    sessions.

-   **PD-DBS longitudinal rats (real data, no ground-truth plasticity
    rule)**: Plasticity inference is evaluated via agreement with
    independently estimated slow biomarkers: (i) trajectory alignment
    between inferred motif coefficients $\mathbf{c}$ and empirical
    functional connectivity (FC) across weeks, by PCA-embedding both and
    using Procrustes alignment, and (ii) group-level effects using the
    magnitude of plasticity change (e.g., $\\|\Delta c\\|_2$) over the DBS
    interval.

## References

\[1\] Borra, F., Cocco, S., & Monasson, R. (2024). Task learning through
stimulation-induced plasticity in neural networks. PRX Life, 2(4),
043014.

---

### Meta-Review · Area_Chair_Qakg · 2025-12-11

**Summary:**

The reviewers expressed significant concerns regarding the evaluation rigor and the identifiability of the inferred dynamics. A major critique was the lack of comprehensive baselines, with one reviewer noting that comparing only against MD-SSM was insufficient and another requesting benchmarks against other meta-learning approaches mentioned in the introduction. Furthermore, the validity of the recovered plasticity rules was questioned; reviewers pointed to visual mismatches between ground truth and inferred weights in the figures and raised theoretical doubts about whether the framework guarantees a unique separation of fast and slow dynamics. There were specific calls for better metrics, such as a baseline for the DSA score and direct quantitative evaluation of the inferred plasticity rules rather than relying solely on state reconstruction error.

Presentation and technical clarity were also recurring issues. Reviewers found some figures too small or qualitatively inconsistent and criticized the placement of important equations in the appendix. There was ambiguity regarding key definitions, such as what constitutes a session, and a lack of justification for specific design choices like the synthetic plasticity rules used in experiments. Finally, the reviewers requested more detailed analyses of hyperparameter sensitivity, questioning how choices regarding rank, regularization, and learning rates impact the results and whether the method effectively scales to larger networks without encoding connectivity information in input or readout weights.

**Reviewer Concerns:**

The authors attempted to address the following concerns:

- Need for other benchmarks: The authors added a closed-loop stimulation-evoked reconfiguration dataset to test if the model can recover connectivity changes driven by adaptive inputs.

- Need for better metrics for plasticity: The authors used DSA and trajectory alignment to quantify the slow-timescale evolution directly, moving beyond simple state prediction.

- Need for other baselines/controls: Implemented shuffle controls to establish chance baselines and added comparisons against MD-SSM and Hierarchical PLRNN to demonstrate superior plasticity forecasting.

- Presentation: Clarified that the model recovers latent reorganization directions rather than exact weights and improved figure clarity with error bars.

**Reviewer Scores:**

Given the extensive responses from the authors, and their attempts to address the concerns, I suspect that the scores would have gone from 4,4,6,4, to 5,5,6-7,5.

---

### Decision · Program_Chairs · 2026-01-26

Accept (Poster)